EMBO
Molecular Medicine

# Myeloid p38α signaling promotes intestinal IGF-1 production and inflammation-associated tumorigenesis

Catrin Youssif[1], Monica Cubillos-Rojas[1], Mònica Comalada[1], Elisabeth Llonch[1], Cristian Perna[2], Nabil Djouder[3] & Angel R Nebreda[1,4,*] (iD)

## Abstract

The protein kinase p38α plays a key role in cell homeostasis, and p38α signaling in intestinal epithelial cells protects against colitis-induced tumorigenesis. However, little is known on the contribution of p38α signaling in intestinal stromal cells. Here, we show that myeloid cell-specific downregulation of p38α protects mice against inflammation-associated colon tumorigenesis. The reduced tumorigenesis correlates with impaired detection in the colon of crucial chemokines for immune cell recruitment. We identify insulin-like growth factor-1 (IGF-1) as a novel mediator of the p38α pathway in macrophages. Moreover, using genetic and pharmacological approaches, we confirm the implication of IGF-1 produced by myeloid cells in colon inflammation and tumorigenesis. We also show a correlation between IGF-1 pathway activation and the infiltration of myeloid cells with active p38α in colon samples from patients with ulcerative colitis or colon cancer. Altogether, our results uncover an important role for myeloid IGF-1 downstream of p38α in colitis-associated tumorigenesis and suggest the interest in evaluating IGF-1 therapies for inflammation-associated intestinal diseases, taking into consideration IGF-1 signaling and immune cell infiltration in patient biopsies.

**Keywords** colon tumorigenesis; IGF-1; intestinal inflammation; macrophage; p38 MAPK

**Subject Categories** Cancer; Digestive System; Immunology

## Introduction

Colorectal cancer (CRC) is the third most common cancer and the fourth leading cause of cancer-related mortality. The risk of developing CRC is significantly increased in individuals diagnosed with inflammatory bowel disease (IBD), a chronically relapsing inflammatory disorder (Coussens & Werb, 2002). The link between inflammation and carcinogenesis was postulated more than 150 years ago (Balkwill & Mantovani, 2001). However, in spite of the progress in IBD research, the mechanisms and the pathogenesis underlying colitis-associated cancer (CAC) are only partially understood (Kaplan & Ng, 2017).

The co-evolution of mammals with their intestinal flora has led to a situation of tolerance in the gut. The equilibrium between immune activation and suppression (intestinal tolerance) is marked by a situation of controlled "physiological inflammation", where distinct populations of resident and inflammatory macrophages in the gut maintain a balance and ensure protective immunity when required (Fiocchi, 2008; Mowat & Bain, 2011). Deregulation of this equilibrium, as it happens in IBD, implicates an imbalance between pro- and anti-inflammatory cytokines, impeding the resolution of inflammation and leading to disease perpetuation and tissue destruction (Neurath, 2014). Cytokines play an important role in the repair of the intestinal epithelia, but have been also implicated in tumor promotion (Schneider et al, 2017).

Several signaling pathways are involved in macrophage activation, phenotype plasticity, and the regulation of proliferation and survival. One of these pathways relies on the protein kinase p38α (Cuadrado & Nebreda, 2010), which regulates cytokine production and inflammatory responses (Kim et al, 2008; Wagner & Nebreda, 2009). Genetic inactivation of p38α in intestinal epithelial cells has provided evidence for the ability of this signaling pathway to suppress inflammation-associated colon tumorigenesis (Gupta et al, 2014). On the other hand, mice with myeloid cell-specific deletion of p38α show less inflammation in response to dextran sodium sulfate (DSS) when compared to wild-type (WT) mice (Otsuka et al, 2010). However, whether and how myeloid p38α signaling can affect intestinal repair mechanisms and tumorigenesis remains unclear. During colorectal carcinogenesis, colonic epithelial cells accumulate genetic mutations, which are mostly induced by environmental factors and confer a selective growth advantage.

1   Institute for Research in Biomedicine (IRB Barcelona), Barcelona Institute of Science and Technology, Barcelona, Spain
2   Hospital Universitario Ramón y Cajal, IRYCIS, Madrid, Spain
3   Centro Nacional de Investigaciones Oncológicas (CNIO), Madrid, Spain
4   ICREA, Barcelona, Spain
    *Corresponding author. Tel: +34 934031379; E-mail: angel.nebreda@irbbarcelona.org

Nevertheless, it is clear that an inflammatory microenvironment involving growth factors and cytokines secreted by activated monocytes and macrophages plays a pivotal role in the formation of tumors (Baylin & Ohm, 2006; Chanmee et al, 2014). Therefore, targeting the recruitment of immune cells to the site of inflammation has emerged as a potential strategy to impede tumorigenesis (Grivennikov et al, 2010).

Myeloid cells are the predominant leukocytes that infiltrate tumors and are known to support tumor initiation and progression. Since p38α controls the production of leukocyte chemo-attractants and other pro-inflammatory mediators (Kim et al, 2008; Cuadrado & Nebreda, 2010), we investigated the contribution of myeloid p38α signaling to CAC. We found that mice with p38α-deficient myeloid cells show reduced inflammation-associated colon tumorigenesis. In a recent screening for bone marrow-derived macrophage (BMDM)-produced cytokines that are regulated by p38α signaling, we identified IGF-1, a peptidic hormone that is involved in inflammatory cell recruitment and has repair functions with high tumorigenic potential (Mourkioti & Rosenthal, 2005; Roussos et al, 2011; Tonkin et al, 2015). We show that p38α signaling regulates IGF-1 expression in intestinal macrophages and that IGF-1 signaling promotes colitis and inflammation-associated colon tumorigenesis. Our results indicate that myeloid p38α through the production of IGF-1 controls colon inflammation and tumorigenesis.

# Results

## Mice with p38α-deficient myeloid cells show decreased susceptibility to colon tumorigenesis

To evaluate the role of myeloid p38α in inflammation-associated colon tumorigenesis, we generated mice expressing LysM-Cre and p38α-lox alleles (p38α-$\Delta^{MC}$). Intestinal macrophages were isolated (Fig EV1A), and the efficiency of p38α downregulation was confirmed by qRT–PCR quantification of the floxed exon 2 in p38α mRNA (Fig 1A) and by immunoblotting of p38α in peritoneal macrophages (Fig EV1B). We found that WT mice treated with azoxymethane (AOM) and DSS to induce CAC (Appendix Fig S1A) developed rectal prolapse, which was observed less often in p38α-$\Delta^{MC}$ mice (Appendix Fig S1B). Macroscopic tumors were mainly located in the distal to middle colon of both WT and p38α-$\Delta^{MC}$ mice (Fig 1B). The overall tumor load was significantly reduced in p38α-$\Delta^{MC}$ mice compared to WT mice, mainly due to a decrease in the number of tumors (Fig 1C). Although the average tumor size was

similar (Fig EV1C), p38α-$\Delta^{MC}$ mice had less tumors larger than 4 mm and no tumors larger than 6 mm (Fig EV1D). In agreement with the reduced tumorigenesis and the lower number of big tumors observed in p38α-$\Delta^{MC}$ mice, we also found lower cell proliferation rates evaluated by Ki67 staining in tumors from these mice (Fig EV1E). However, apoptosis determined by TUNEL or cleaved caspase-3 staining did not show significant differences between tumors from WT and p38α-$\Delta^{MC}$ mice (Appendix Fig S1C and D).

## Myeloid p38α controls the tumor-promoting inflammatory microenvironment

Given the important contribution of immune cells to the inflammatory microenvironment, we evaluated the number of inflammatory monocytes in the bone marrow. The C-C chemokine receptor type (CCR) 2 is very important for Ly6C$^{hi}$ monocyte trafficking, and it is well accepted that Ly6C$^{hi}$ monocytes rely on CCR2 to egress from the bone marrow to the inflamed and healthy intestine, where they can give rise to different types of macrophages (Bain & Mowat, 2014). We found significantly less Ly6C$^{hi}$CCR2$^+$ inflammatory monocytes in the bone marrow of p38α-$\Delta^{MC}$ mice compared to WT mice, indicating a weaker inflammatory response in tumor-bearing p38α-$\Delta^{MC}$ mice (Fig 1D and Appendix Fig S1E). Therefore, we evaluated the immune cell infiltrate in the tumors. In agreement with the reduced levels of inflammatory monocytes detected in the bone marrow of p38α-$\Delta^{MC}$ mice, tumors in these mice showed less macrophage (F4/80$^+$) infiltration compared to the those in WT mice (Fig 1E and Appendix Fig S1F). We further evaluated the phosphorylation status of signal transducer and activator of transcription 3 (STAT3), a potent activator of inflammatory pathways that contributes to oncogenic signaling leading to enhanced cell proliferation and tumor growth (Yu et al, 2009; Sanchez-Lopez et al, 2016). As expected, colon tumors showed enhanced STAT3 phosphorylation compared to the normal epithelium, and interestingly, STAT3 phosphorylation was reduced in tumors from p38α-$\Delta^{MC}$ mice compared to WT mice (Fig EV1F). Next, we evaluated the expression of cytokines and chemokines in tumors (Fig 1F and Appendix Fig S1G). In line with the decreased macrophage infiltration and STAT3 phosphorylation levels observed in tumors from p38α-$\Delta^{MC}$ mice, these tumors also showed reduced levels of chemokines important for monocyte/macrophage recruitment such as CXCL10, CCL3, and CCL2 (Balkwill & Mantovani, 2001; Zimmerman et al, 2008; Mowat & Bain, 2011), as well as STAT3-activating cytokines, such as G-CSF and IL-27 (Rebe et al, 2013).

---

**Figure 1. Downregulation of p38α in myeloid cells reduces colitis-associated tumorigenesis.**

A   Analysis by qRT–PCR of the levels of floxed exon 2 versus exon 12 (as a control) of the mRNA encoding p38α in intestinal macrophages ($n \geq 5$).
B   Representative images of colon tumors in AOM/DSS-treated mice. Red arrows indicate macroscopically visible tumors that were measured.
C   Average tumor number and load in AOM/DSS-treated mice ($n \geq 11$). The experiment was performed three times.
D   Percentage of Ly6C$^{hi}$CCR2$^+$ cells in the bone marrow cells that were alive and CD45$^+$CD11b$^+$ from AOM/DSS-treated mice ($n \geq 3$).
E   Representative sections from normal colon epithelia and colon tumors stained for F4/80. Quantifications are shown in the histogram ($n = 4$). The area indicated was used for quantification and was selected for all the immunohistochemistry quantifications of epithelium. Scale bars, 100 μm.
F   Quantifications of a mouse cytokine antibody array that was interrogated using pools of 3-mm tumors derived from AOM/DSS-treated mice either WT or p38α-$\Delta^{MC}$ ($n = 8$/genotype). Arbitrary units (a.u.) are referred to the expression level of each cytokine in tumors from WT mice, which was given the value of 1.

Data information: Statistical analysis was performed by using Mann–Whitney test for the comparison of two groups or ANOVA using Bonferroni post hoc correction for multiple groups. Data are expressed as the average ± SD.

---

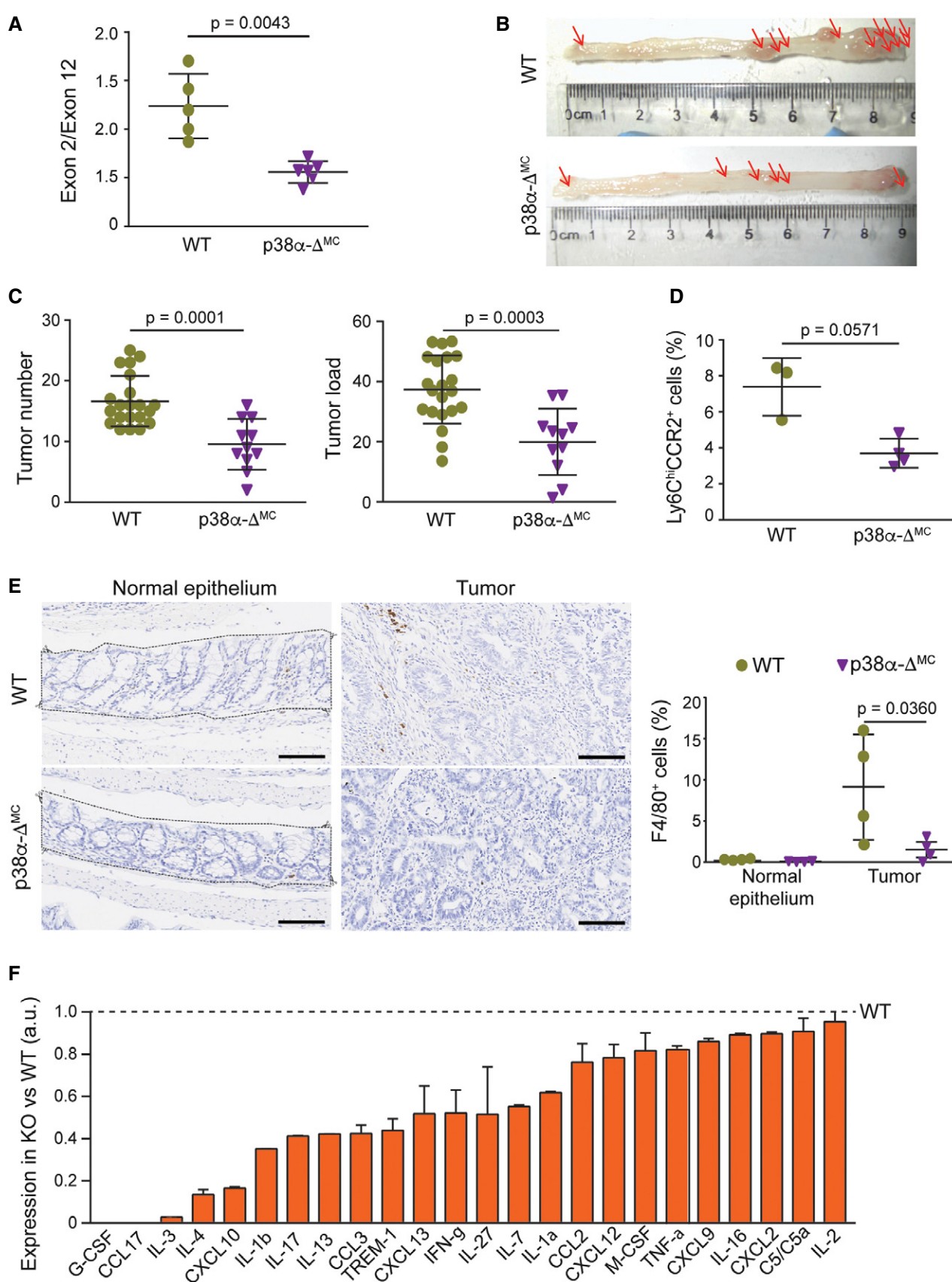

**Figure 1.**

Since inflammation is an essential component of the AOM/DSS tumorigenesis protocol, we investigated the contribution of myeloid p38α to the DSS response (Appendix Fig S2A). In agreement with a previous report (Otsuka et al, 2010), we found that p38α-$\Delta^{MC}$ mice were less susceptible to DSS-induced colitis, as indicated by reduced body weight loss (Appendix Fig S2B) and a lower disease activity index (DAI; Fig 2A). In line with the decreased severity of colitis indicators, the large areas of complete crypt loss and erosions observed in colons from DSS-treated WT mice were strongly reduced in p38α-$\Delta^{MC}$ mice. However, WT and p38α-$\Delta^{MC}$ mice showed no differences in colon histology prior to DSS administration (Fig 2B and EV2A). We also observed that the numbers of macrophages (F4/80$^{+}$), leukocytes (CD45$^{+}$), activated neutrophils (MPO$^{+}$), and T cells (CD3$^{+}$) were all reduced in the immune cell infiltrates of colons from DSS-treated p38α-$\Delta^{MC}$ mice compared to WT mice (Fig 2C and EV2B–D, and Appendix Fig S2C–E).

Infiltrating immune cells produce cytokines that activate STAT3 and its target genes contributing to tumor-promoting inflammation (Yu et al, 2009). Accordingly, STAT3 phosphorylation was reduced in the colon epithelial cells of DSS-treated p38α-$\Delta^{MC}$ mice compared to WT mice (Fig 2D). Moreover, ELISA analysis showed that the inflammatory mediator IL-1β was downregulated in colons from DSS-treated p38α-$\Delta^{MC}$ mice compared to WT mice, and TNF-α was also induced at lower levels upon DSS treatment in p38α-$\Delta^{MC}$ mice (Appendix Fig S2F). Further analysis of colons at day 7 after DSS treatment showed the downregulation of several other pro-inflammatory cytokines and chemokines in p38α-$\Delta^{MC}$ mice, which likely contribute to inflammatory cell recruitment and disease promotion (Fig 2E and Appendix Fig S2G).

### Downregulation of p38α in myeloid cells reduces IGF-1 signaling during intestinal inflammation and tumorigenesis

We have identified IGF-1 as one extracellular factor potentially regulated by p38α signaling in myeloid cells. Both intracellular and extracellular IGF-1 protein levels were reduced in p38α-deficient BMDMs compared with WT BMDMs (Fig 3A). IGF-1 mRNA expression was more potently induced by IL-4 than by LPS (Fig 3B), in agreement with its proposed role as marker for wound-healing macrophages (Tonkin et al, 2015; Spadaro et al, 2017), which contribute to tumor progression (Murray & Wynn, 2011). The implication of p38α signaling in IGF-1 expression by macrophages was confirmed by using chemical inhibitors (Fig 3B and C).

To confirm that p38α downregulation in myeloid cells affects IGF-1 signaling during inflammation and tumorigenesis, we analyzed IGF-1 levels in mice treated with DSS or AOM/DSS. In response to DSS, intestinal macrophages switch from the initial classical activation phenotype to a wound-healing phenotype in the repair phase. Accordingly, we detected a clear reduction in IGF-1 levels in colons from p38α-$\Delta^{MC}$ mice compared to WT mice during the repair phase at day 13, whereas no significant differences were observed in untreated colons or during the acute inflammatory phase at day 7 (Fig 4A). Analysis by qRT–PCR also showed lower levels of IGF-1 mRNA at day 13 in colon extracts from p38α-$\Delta^{MC}$ mice compared to WT mice (Appendix Fig S3A). Consistently, IGF-1 mRNA levels were also reduced in p38α-deficient intestinal macrophages compared to WT macrophages at day 13 (Fig 4B), and the differences were even clearer than in whole colon extracts. Taken together, our results support a key role for p38α signaling in IGF-1 production by myeloid cells during the repair phase in the inflamed colon. However, we observed no differences in serum IGF-1 levels between WT and p38α-$\Delta^{MC}$ mice (Appendix Fig S3B), suggesting that changes in IGF-1 signaling in the intestines were probably produced locally by myeloid cells.

Consistent with the known mitogenic properties of IGF-1, we found significant differences in cell proliferation as determined by Ki67 staining between the colons of DSS-treated WT and p38α-$\Delta^{MC}$ in the repair phase at day 13 (Fig 4C and Appendix Fig S3C), when IGF-1 protein levels were significantly different in colon extracts (see Fig 4A above). IGF-1 binds to and induces the autophosphorylation of the IGF-1 receptor (IGF1R). To assess IGF-1 signaling activity during intestinal inflammation and tumorigenesis, we performed immunohistochemistry staining for phospho-IGF1R followed by TMarker analysis (Schuffler et al, 2013) to differentiate between different staining intensities (Appendix Fig S3D). Importantly, IGF1R phosphorylation was reduced in colon tumors from p38α-$\Delta^{MC}$ mice compared to WT mice (Fig 4D).

### IGF-1 promotes intestinal inflammation and inflammation-associated tumorigenesis

IGF-1 has been implicated in inflammatory processes and immune cell recruitment (Mourkioti & Rosenthal, 2005). To evaluate the role of myeloid IGF-1 in intestinal inflammation, we used mice expressing LysM-Cre and IGF-1-lox alleles (IGF-1-$\Delta^{MC}$). The efficiency of IGF-1 mRNA downregulation was confirmed by qRT–PCR in isolated intestinal macrophages (Fig 5A) and peritoneal macrophages (Fig EV3A). We observed that DSS-treated IGF-1-$\Delta^{MC}$ mice showed

---

**Figure 2.  Myeloid deletion of p38α decreases colitis susceptibility and leukocyte recruitment to the inflamed intestine.**

A   Disease activity index was recorded daily in DSS-treated mice ($n \geq 9$). The experiment was performed three times.

B   Epithelial damage was quantified in untreated mice or mice treated with DSS for 6 days and analyzed at the indicated times (days) using H&E-stained colon sections ($n \geq 5$).

C   Representative colon sections from untreated mice or mice treated with DSS for 6 days were analyzed at day 7 for F4/80 staining. Quantifications are shown in the histogram ($n \geq 4$). Scale bars, 100 μm.

D   Representative colon sections from untreated mice or mice treated with DSS for 6 days and analyzed at day 7 were stained for phospho-STAT3. Quantifications are shown in the histogram ($n \geq 4$). Scale bars, 100 μm.

E   Quantifications of a mouse cytokine antibody array that was interrogated using pools of whole colon extracts derived from DSS-treated mice either WT or p38α-$\Delta^{MC}$ at day 7 ($n = 8$/genotype). Arbitrary units (a.u.) are referred to the expression level of each cytokine in WT mice, which was given the value of 1.

Data information: Statistical analysis was performed by ANOVA using Bonferroni post hoc correction. Data are expressed as the average ± SD.

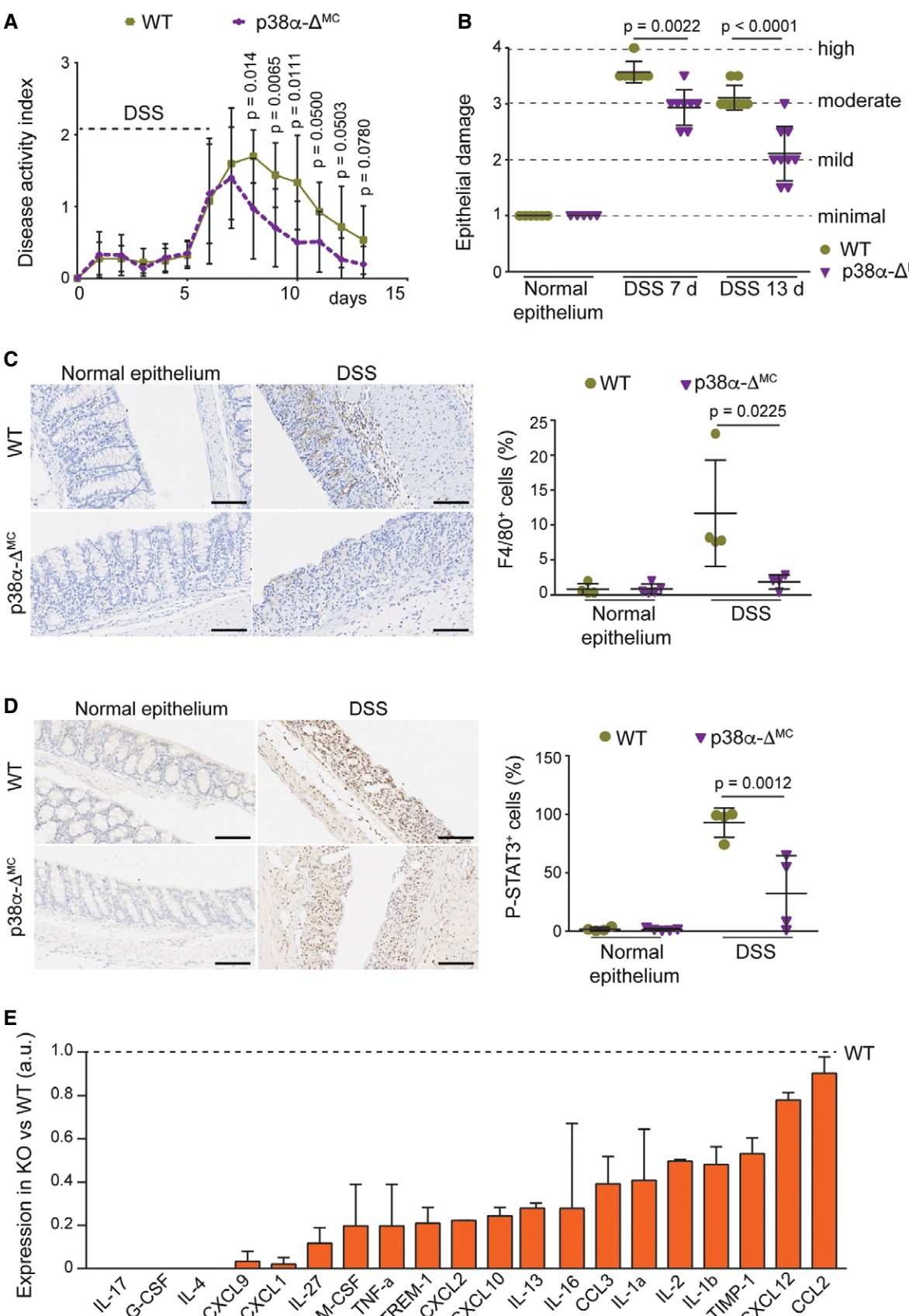

Figure 2.

small differences in body weight loss (Fig EV3B), but had a more significant reduction in DAI (Fig EV3C and Appendix Fig S3E) as well as less epithelial damage at day 7 compared to WT mice (Figs 5B and EV3D).

Next, we investigated the role of IGF-1 in inflammatory cell recruitment and found significantly less Ly6C$^{hi}$CCR2$^{+}$ inflammatory monocytes in the bone marrow of DSS-treated IGF-1-$\Delta^{MC}$ mice compared to WT mice (Fig 5C). In agreement with this, we observed a reduced number of macrophages (F4/80$^{+}$; Fig 5D), and STAT3 phosphorylation levels were also reduced in colons from DSS-treated IGF-1-$\Delta^{MC}$ mice (Fig EV3E). Interestingly, IGF-1 mRNA levels were reduced in colons from untreated IGF-1-$\Delta^{MC}$ mice compared to WT mice, suggesting an important contribution of the IGF-1 produced by myeloid cells in the colon (Fig 5E). In line with this idea, quantification of IGF1R phosphorylation revealed a reduction of IGF-1 signaling in colons from DSS-treated IGF-1-$\Delta^{MC}$ mice compared to WT mice (Fig EV3F).

To confirm the implication of IGF-1 signaling in DSS-induced intestinal inflammation, we used PQ401, a chemical inhibitor of IGF1R autophosphorylation. WT mice were treated daily with PQ401, starting 1 day before the DSS treatment, and were compared with WT and p38α-$\Delta^{MC}$ mice treated with vehicle and DSS (Appendix Fig S3F). As expected, the levels of IGF1R phosphorylation were reduced in the colon of mice treated with PQ401 (Appendix Fig S3G). In agreement with our results using IGF-1-$\Delta^{MC}$ mice, treatment of WT mice with PQ401 reduced DSS-induced body weight loss, DAI, and epithelial damage closer to the values observed in p38α-$\Delta^{MC}$ mice (Fig 5F and Appendix Fig S4A–C). Interestingly, inflammatory monocytes were reduced in the bone marrow of PQ401-treated WT mice, as observed in p38α-$\Delta^{MC}$ mice (Fig 5G). Consistently, treatment with PQ401 also reduced the number of macrophages infiltrating the colon of DSS-treated WT mice to similar levels as in p38α-$\Delta^{MC}$ mice (Fig 5H and Appendix Fig S4D). Of note, the treatment of p38α-$\Delta^{MC}$ mice with PQ401 did not further ameliorate DSS-induced DAI and epithelial damage or the number of inflammatory monocytes in the bone marrow (Appendix Fig S4E–G). The IGF-1 contribution to the DSS-induced phenotypes was further supported by treating p38α-$\Delta^{MC}$ mice with recombinant IGF-1. We found that in response to DSS, p38α-$\Delta^{MC}$ mice treated with IGF-1 showed a DAI, number of pro-inflammatory monocytes in the bone marrow, epithelial damage, and STAT3 phosphorylation levels more similar to WT mice than to p38α-$\Delta^{MC}$ mice (Fig 6A–D). Taken together, these results support that IGF-1 signaling contributes to DSS-induced inflammation and that IGF-1 produced by myeloid cells plays a key role in the process.

To address the importance of IGF-1 in inflammation-associated tumorigenesis, we treated WT and IGF-1-$\Delta^{MC}$ mice with the AOM/DSS protocol. Similar to p38α-$\Delta^{MC}$ mice, IGF-1-$\Delta^{MC}$ mice showed a decreased number of colon tumors compared to WT mice (Fig 7A), with less tumors larger than 4 mm and no tumors larger than 6 mm (Fig 7B). Of note, colon tumors from IGF-1-$\Delta^{MC}$ mice exhibited reduced macrophage infiltration compared to those from WT mice (Fig 7C), further suggesting the implication of IGF-1 in immune cell recruitment. To evaluate the potential therapeutic interest of targeting IGF-1 signaling in colorectal tumorigenesis, WT mice were treated with the AOM/DSS protocol and then were split in two groups, one received PQ401 and the other one vehicle for 20 days (Fig 7D). The results indicated that inhibition of IGF-1 signaling reduced the colon tumor numbers in WT mice to similar levels as in p38α-$\Delta^{MC}$ mice (Fig 7E).

## Myeloid p38α controls inflammatory cell recruitment to the colon through the regulation of chemokines

Chemokines are important modulators of inflammatory processes, and inappropriate chemokine expression can cause a massive and destructive leukocyte infiltration (Balkwill & Mantovani, 2001). Under natural conditions, the gut is marked by a situation of controlled "physiological inflammation", due to the constant exposure to environmental microbiota (Mowat & Bain, 2011). Indeed, resembling a physiological inflammatory state, p38α-$\Delta^{MC}$ mice under homeostatic conditions showed decreased amounts of inflammatory monocytes in the bone marrow compared to WT mice (Fig EV4A). Besides its role as a growth hormone, IGF-1 can function as chemo-attractant (Roussos et al, 2011). Accordingly, monocyte recruitment was reduced in the bone marrow of untreated IGF-1-$\Delta^{MC}$ mice compared to WT mice (Fig EV4B), as observed upon induction of colitis with DSS in IGF-1-$\Delta^{MC}$ mice and in PQ401-treated WT mice (see above, Fig 5C and G).

We investigated the chemokine levels in colons from untreated WT and p38α-$\Delta^{MC}$ mice to evaluate the effect of p38α downregulation on chemo-attractants other than IGF-1. Interestingly, we observed an overall downregulation of chemokines in p38α-$\Delta^{MC}$ mice compared to WT mice, consistent with the reduced recruitment of monocytes/macrophages (Fig EV4C). We confirmed the downregulation of Chemerin, IL-16, CCL2, and CCL12 mRNAs in isolated intestinal macrophages from p38α-$\Delta^{MC}$ mice. However, in IGF-1-$\Delta^{MC}$ mice only CCL2 mRNA tends to be less expressed (Appendix Fig S5A). In addition, we analyzed the expression of these chemokines in whole colons and observed significant downregulation of IL-16 mRNA in p38α-$\Delta^{MC}$ mice and of CCL-9 in IGF-1-$\Delta^{MC}$ mice compared

---

**Figure 3.  p38α regulates IGF-1 production by macrophages.**

A   Whole protein lysates (left panel) or supernatants (right panel) from bone marrow-derived macrophages (BMDMs) were used to analyze IGF-1 protein levels by ELISA ($n = 5$).

B   BMDMs were starved for 18 h prior to stimulation with either LPS (10 ng/ml) or IL-4 (10 ng/ml) for the indicated times in the presence or absence of the indicated p38α inhibitors or vehicle (DMSO). The inhibitors were added to the medium 1 h prior to stimulation with LPS or IL-4, and IGF-1 mRNA expression levels were measured by qRT–PCR ($n = 3$). Arbitrary units (a.u.) are referred to the expression level in DMSO-treated control at 3 h, which was given the value of 1.

C   BMDMs were starved overnight, and the indicated p38α inhibitors or DMSO were added to the medium 1 h prior to stimulation with IL-4 for 6 h. Supernatants were collected and used to measure IGF-1 protein levels by ELISA ($n = 3$). The expression level in the DMSO-treated control (about 700 pg/ml) was given the value of 1.

Data information: Statistical analysis was performed by using Mann–Whitney test for the comparison of two groups or ANOVA using Bonferroni post hoc correction for multiple groups. Data are expressed as the average ± SD.

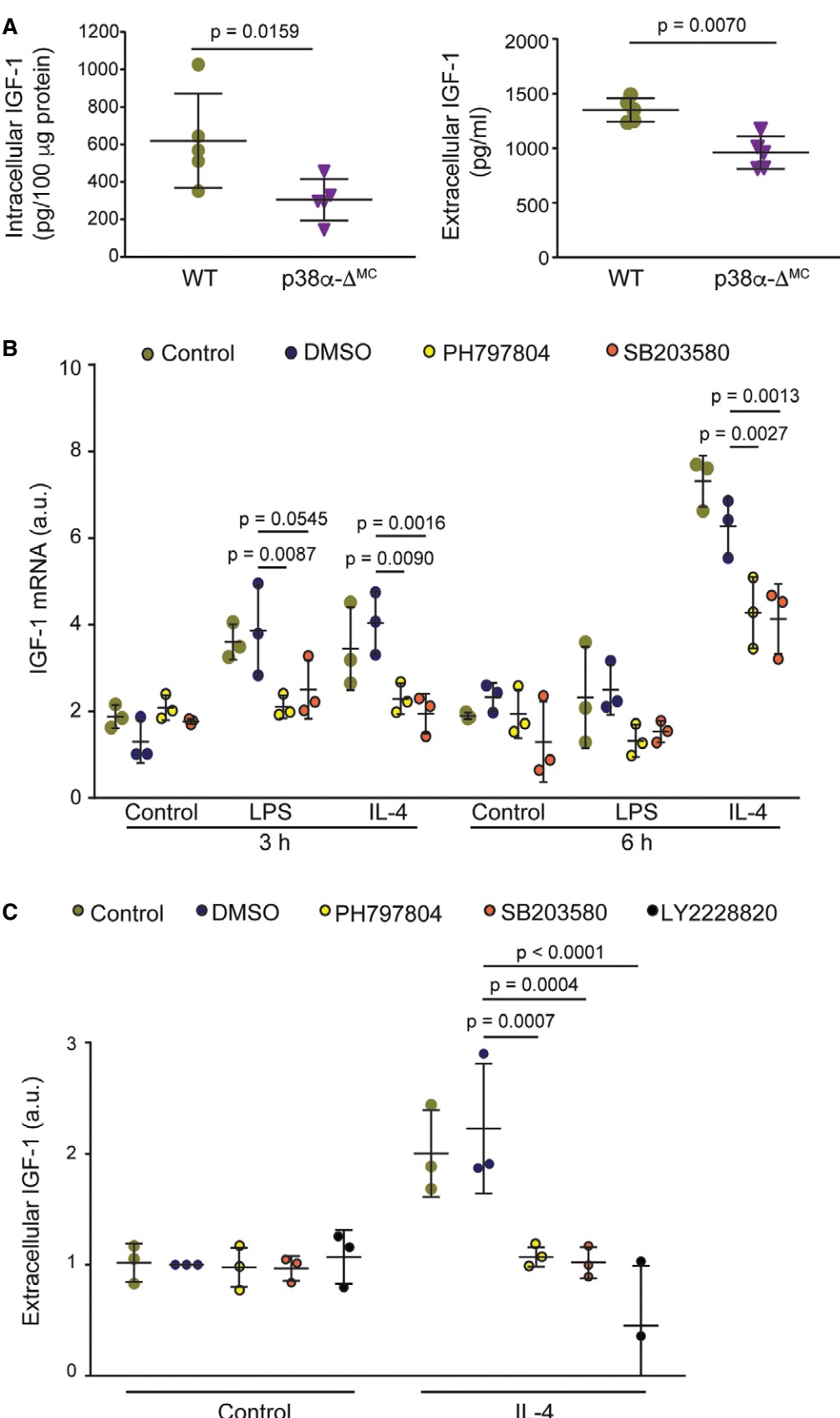

**Figure 3.**

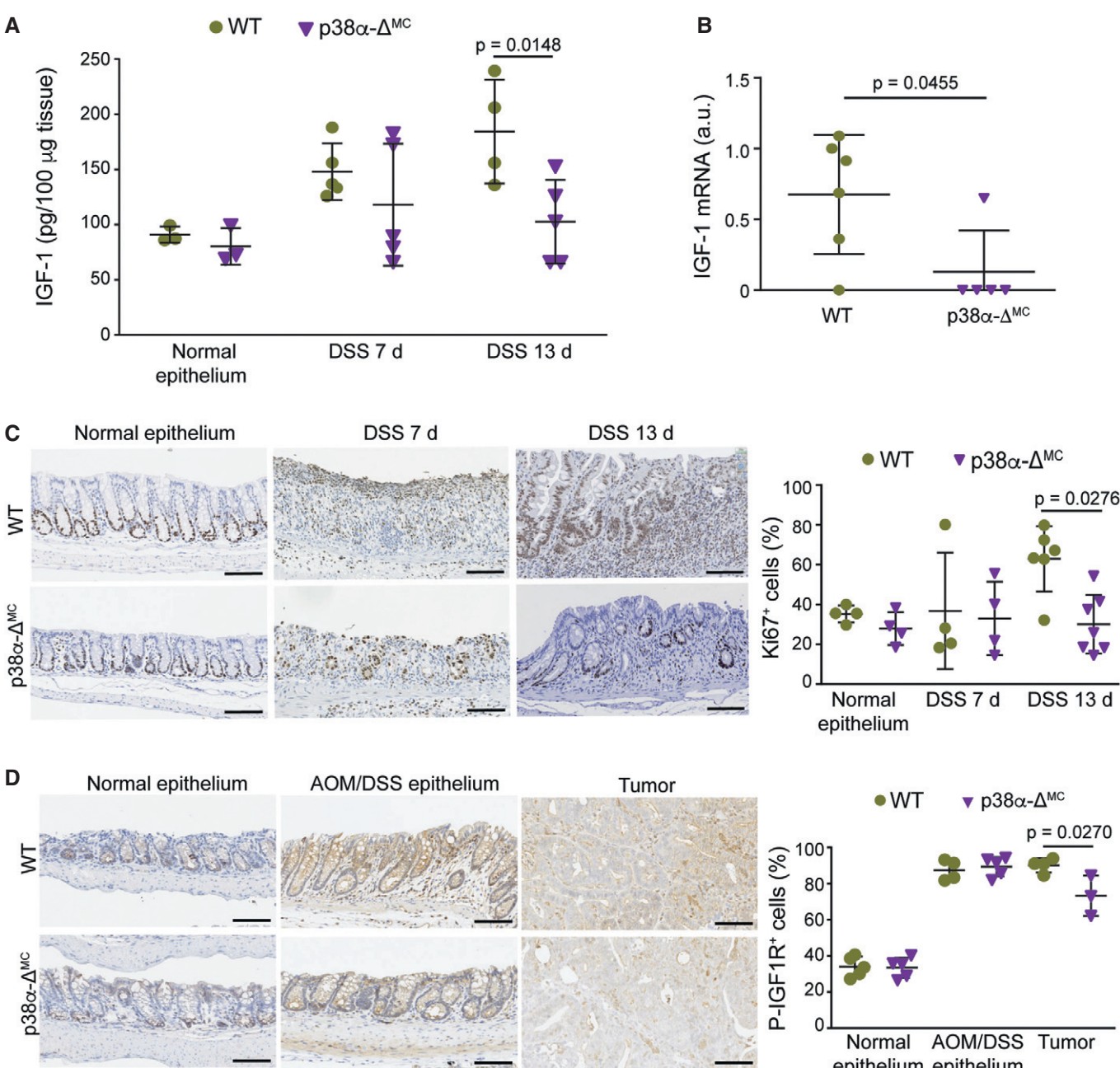

**Figure 4.  Downregulation of myeloid p38α reduces IGF-1 production and signaling during intestinal inflammation and tumorigenesis.**

A  Colon protein lysates obtained from mice either untreated or treated with DSS for 6 days were analyzed at the indicated times to measure IGF-1 protein levels by ELISA ($n \geq 3$).

B  Intestinal macrophages were isolated from mice treated with DSS for 6 days and analyzed at day 13. IGF-1 mRNA levels were quantified by qRT–PCR ($n \geq 5$).

C  Representative colon sections from mice either untreated or treated with DSS for 6 days were analyzed at the indicated times for Ki67 staining. Quantifications are shown in the histogram ($n \geq 4$). Scale bars, 100 μm.

D  Representative colon sections from untreated mice or colon and tumor sections from AOM/DSS-treated mice were stained for phospho-IGF1R. Quantifications are shown in the histogram ($n = 4$). Scale bars, 100 μm.

Data information: Statistical analysis was performed by using Mann–Whitney test for the comparison of two groups or ANOVA using Bonferroni *post hoc* correction for multiple groups. Data are expressed as the average ± SD.

to WT mice (Appendix Fig S5B). We have also identified several chemokines that were downregulated at the protein level in colons from p38α-Δ^MC mice (Fig EV4C) without noticeable changes in

mRNA expression (Appendix Fig S5A and B), suggesting potential regulation by myeloid p38α at the translational or post-translational level. Of note, the chemokine reduction (Fig EV4C) was consistent

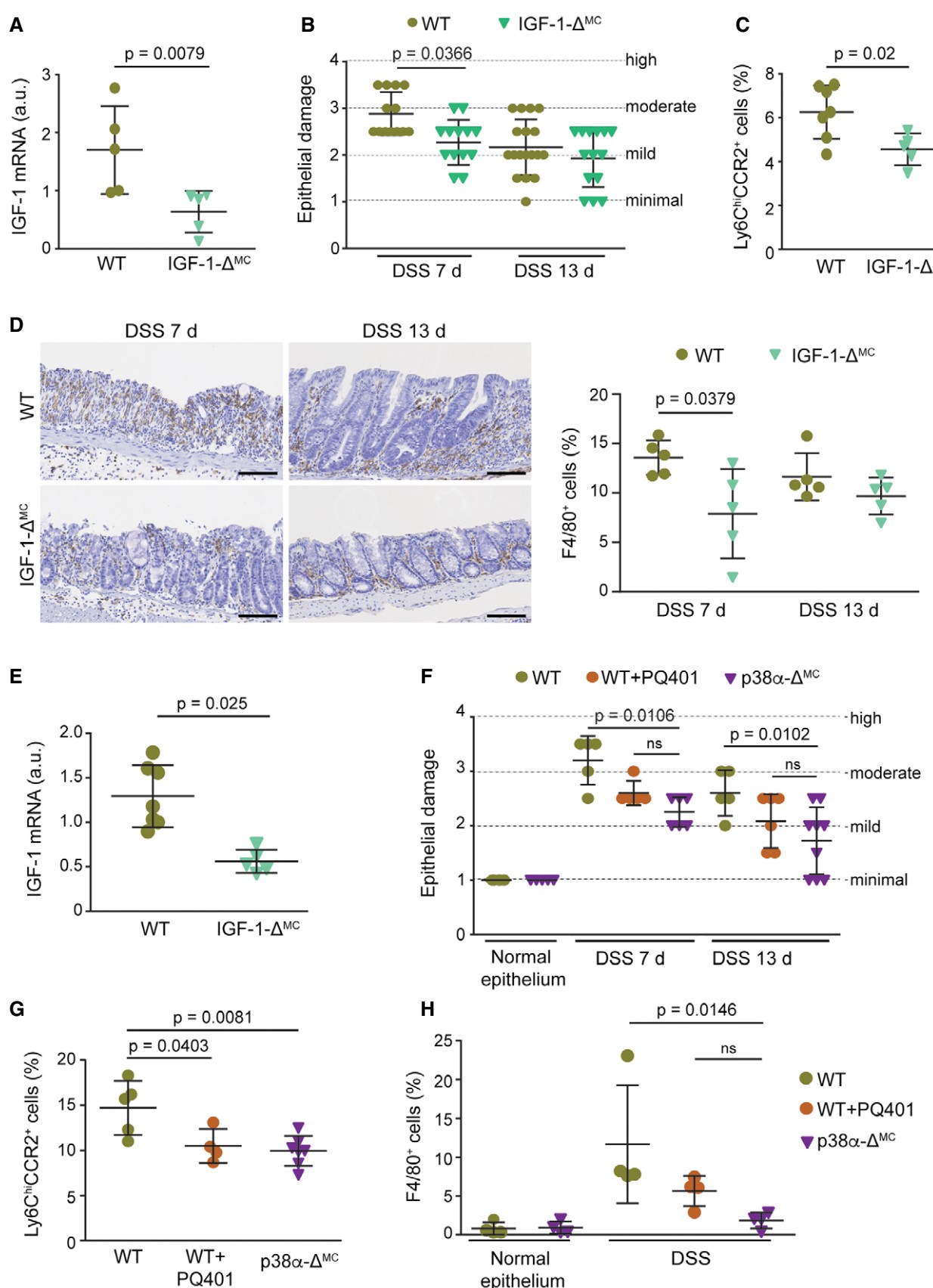

Figure 5.

**Figure 5.  Impaired IGF-1 signaling decreases immune cell recruitment to the inflamed colon.**

A  Relative IGF-1 mRNA levels in isolated intestinal macrophages ($n = 5$).

B  Epithelial damage was evaluated in H&E-stained colon sections from mice treated with DSS for 6 days and analyzed at the indicated days ($n \geq 13$). The experiment was performed three times.

C  Percentage of Ly6C$^{hi}$CCR2$^+$ cells in the bone marrow cells that were alive and CD45$^+$CD11b$^+$ from mice treated for 6 days with DSS and analyzed at day 7 ($n \geq 5$).

D  Representative colon sections from mice treated with DSS for 6 days and analyzed at the indicated days were stained for F4/80. Quantifications are shown in the histogram ($n \geq 4$). Scale bars, 100 μm.

E  IGF-1 mRNA levels in colons from untreated mice were determined by qRT–PCR ($n \geq 5$).

F  Epithelial damage was quantified in mice untreated or treated with either PQ401 or vehicle and then with DSS for 6 days, and analyzed at the indicated days ($n \geq 7$). This experiment was performed twice.

G  Percentage of Ly6C$^{hi}$CCR2$^+$ cells in the bone marrow cells that were alive and CD45$^+$CD11b$^+$ from mice treated with vehicle or PQ401 and DSS for 6 days and sacrificed at day 7 ($n \geq 4$).

H  F4/80-positive cells were quantified in colon sections from untreated mice or mice treated with either PQ401 or vehicle and then untreated or treated with DSS for 6 days, and analyzed at day 7 ($n \geq 4$).

Data information: Statistical analysis was performed by using Mann–Whitney test for the comparison of two groups or ANOVA using Bonferroni *post hoc* correction for multiple. Data are expressed as the average ± SD.

with reduced recruitment of myeloid cells to the colons from p38α-Δ$^{MC}$ mice compared to WT mice (Fig EV4D).

p38α signaling has been linked to the differentiation of several cell types (Cuadrado & Nebreda, 2010), but little is known about the implication of p38α in monocyte/macrophage differentiation. To evaluate whether differences in generation or differentiation of these cells could contribute to the observed phenotypes, we analyzed hematopoietic stem cell populations in the bone marrow of WT and p38α-Δ$^{MC}$ mice but found no significant differences (Appendix Fig S5C). We therefore hypothesized that the differences in inflammatory cell recruitment observed are likely due to reduced chemokine expression, directly or indirectly regulated by myeloid p38α in the colon. Nevertheless, we observed LysM-Cre activity in myeloid cells from the bone marrow, including monocytes and macrophages (Fig EV4E), indicating that disruption of the p38α/IGF-1 axis might affect monocyte recruitment already in the bone marrow. In agreement with this, we detected reduced IGF-1 pathway activation in the bone marrow of p38α-Δ$^{MC}$ mice compared to WT mice (Fig EV4F and G). Taking into consideration that IGF-1-Δ$^{MC}$ mice also exhibited reduced Ly6C$^{hi}$CCR2$^+$ monocytes in the bone marrow under homeostatic conditions (Fig EV4B), our results suggest that the effect on bone marrow monocytes is probably caused directly by IGF-1.

In summary, our results provide evidence that the myeloid p38α/IGF-1 axis facilitates inflammatory cell recruitment in response to DSS-induced colitis and contributes to CAC (Fig EV5). Importantly, we have found a significant correlation between the phosphorylation of p38α in monocytes/macrophages and the phosphorylation of IGF-1 receptor both in colon samples from ulcerative colitis patients and in tumor samples from colon cancer patients (Fig 8A–E), supporting the potential clinical relevance of the p38α/IGF-1 axis in human disease.

## Discussion

Our results demonstrate that myeloid p38α signaling plays a key role in inflammation-associated colon cancer. We show that suppression of p38α in myeloid cells ameliorates the local production of chemo-attractants resulting in reduced inflammatory cell recruitment to the colon and decreased AOM/DSS-induced tumor burden. Tissue-resident and inflammatory macrophages are actively

recruited from circulating bone marrow-derived monocytic precursors, which contribute to modify the tumor microenvironment and facilitate tumor development at various levels (Grivennikov et al, 2010; Chanmee et al, 2014).

We have identified IGF-1 as a novel mediator of p38α signaling in macrophages and showed that IGF-1 produced by myeloid cells contributes to pathological progression of CAC. While p38α is known to regulate various cytokines implicated in inflammation, to our knowledge, this is the first report describing the regulation of IGF-1 by p38α in macrophages. Experiments using a chemical inhibitor have also suggested the implication of the p38α pathway in IGF-1 production by TNF-treated mesenchymal stem cells and adipose progenitor cells in culture (Wang et al, 2006). Our results indicate that p38α regulates IGF-1 production by macrophages *ex vivo* and *in vivo*. Of note, the liver is the major source of circulating IGF-1, but this molecule can be also produced locally and macrophages have been proposed as an important source of extrahepatic IGF-1 (Gow et al, 2010), which is consistent with our results showing downregulation of IGF-1 signaling in the intestines of IGF-1-Δ$^{MC}$ mice. We observed lower IL-4 expression in p38α-Δ$^{MC}$ mice compared to WT mice, both during DSS-induced colitis and in the tumors, and given that IL-4 is a major IGF-1 inducer, it could further contribute to the decreased levels of IGF-1 signaling in p38α-Δ$^{MC}$ mice. Additionally, several studies have identified IL-4 as a major regulator of the phenotypes of tumor-associated macrophages (Wang & Joyce, 2010).

We show that interfering with IGF-1 signaling partially improves epithelial damage and inflammation induced by DSS, which correlates with a reduction of Ly6C$^{hi}$CCR2$^+$ monocytes in the bone marrow. However, DSS-induced epithelial damage in IGF-1-Δ$^{MC}$ mice was not reduced to the same levels as in p38α-Δ$^{MC}$ mice, probably due to the implication of IGF-1 signaling in mucosal repair mechanisms (Zatorski et al, 2016) as well as to the regulation by p38α of other inflammatory mediators (Wagner & Nebreda, 2009; Otsuka et al, 2010), which are not affected by IGF-1 inhibition. Our results suggest that IGF-1 signaling plays a dual role during DSS-induced inflammation. On the one hand, IGF-1 exhibits a clear chemo-attractant function during the acute pro-inflammatory phase, accordingly myeloid cell-specific downregulation of IGF-1 inhibits monocyte and macrophage recruitment reducing epithelial damage induced by DSS. This is consistent with the ability of IGF-1 to mediate chemotaxis of several cell types including tumor cells (Roussos

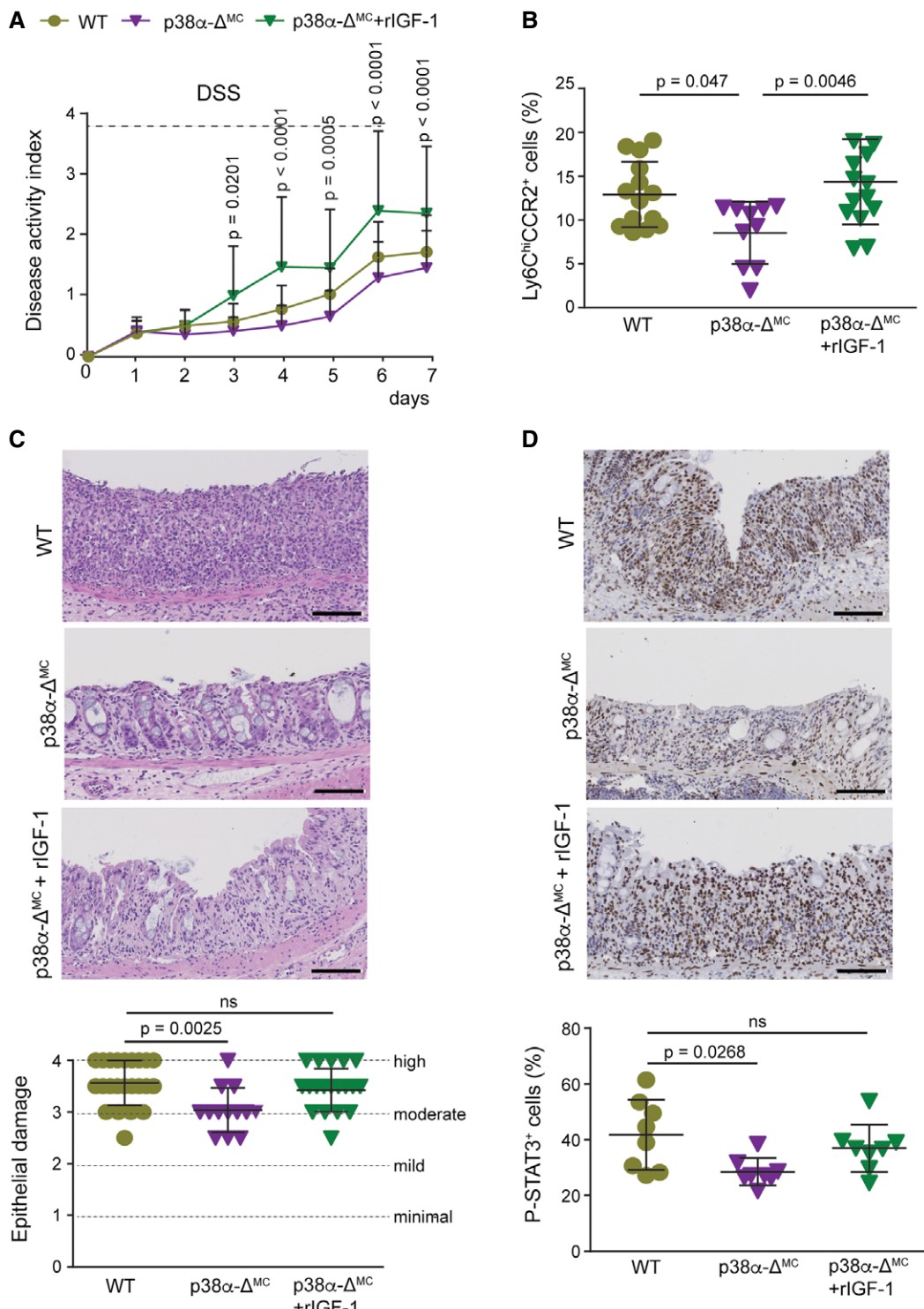

**Figure 6.  IGF-1 promotes inflammatory cell recruitment and DSS-induced epithelial damage.**

A   Disease activity index was recorded daily in DSS-treated mice ($n \geq 13$). This experiment was performed twice.
B   Percentage of Ly6C$^{hi}$CCR2$^+$ cells in the bone marrow cells that were alive and CD45$^+$CD11b$^+$ from mice treated with DSS for 6 days and sacrificed at day 7 ($n \geq 10$).
C   Representative H&E-stained colon sections from mice that were treated with DSS for 6 days and either IGF-1 or PBS and were analyzed at day 7. Quantifications of epithelial damage are shown in the histogram ($n \geq 13$). Scale bars, 100 μm.
D   Representative colon sections from mice that were treated with DSS for 6 days and either IGF-1 or PBS and analyzed at day 7 for phospho-STAT3 staining. Quantifications are shown in the histogram ($n = 8$). Scale bars, 100 μm.

Data information: Statistical analysis was performed by ANOVA using Bonferroni *post hoc* correction. Data are expressed as the average $\pm$ SD.

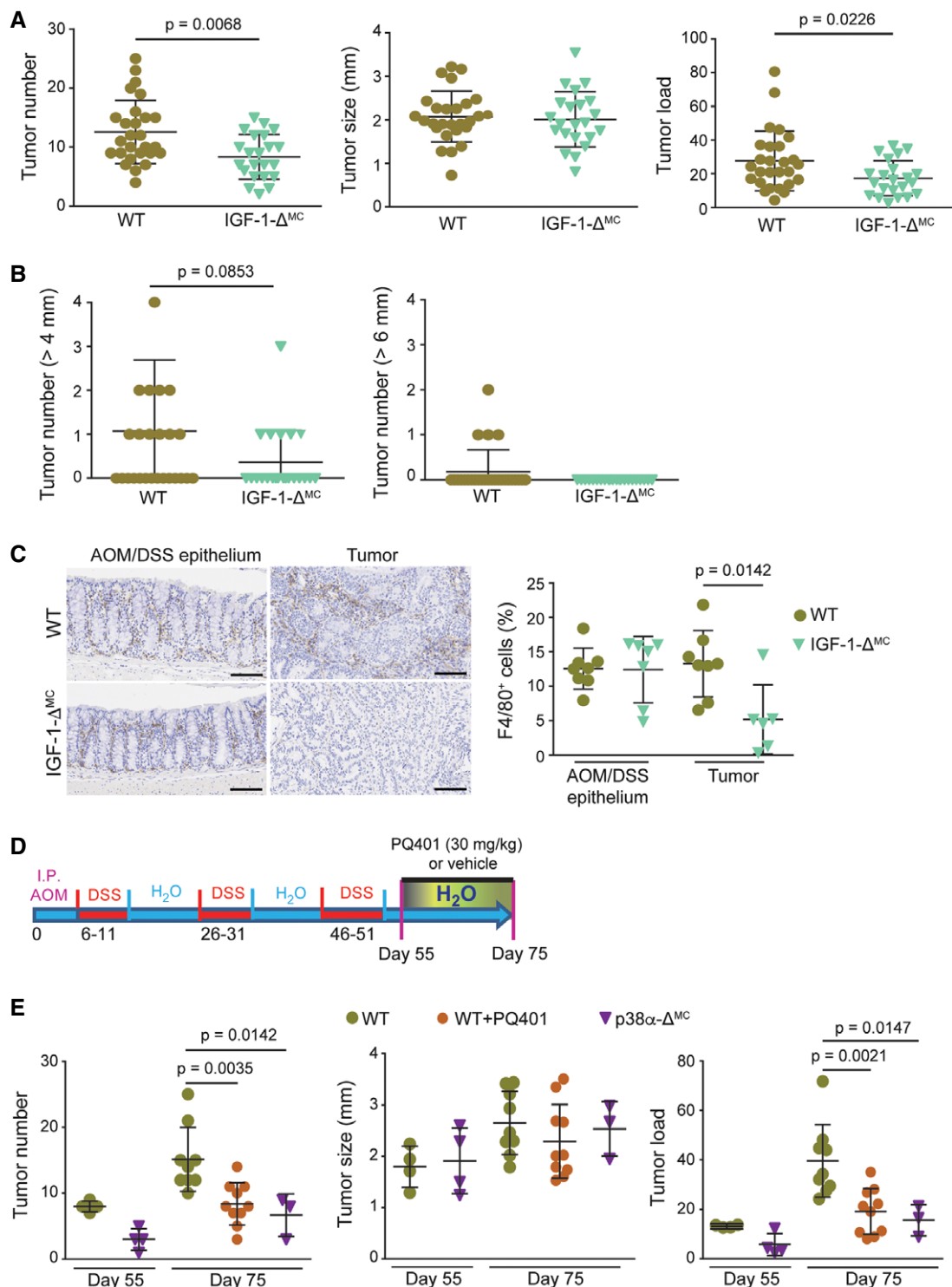

**Figure 7. Inhibition of IGF-1 signaling reduces tumorigenesis induced by AOM/DSS.**

A   Average tumor number, size, and load in AOM/DSS-treated mice ($n \geq 22$). The experiment was performed twice.

B   Number of tumors >4 and >6mm in AOM/DSS-treated mice ($n \geq 22$).

C   Representative sections from AOM/DSS-treated epithelia and colon tumors stained for F4/80. Quantifications are shown in the histogram ($n \geq 6$). Scale bars, 100 μm.

D   Schematic representation of the protocol used for the treatment of mice with AOM/DSS and PQ401 or vehicle. Animals were sacrificed at days 55 and 75.

E   Average tumor number, size, and load in mice treated as described in (D) and analyzed at the indicated days ($n \geq 3$).

Data information: Statistical analysis was performed by using Mann–Whitney test for the comparison of two groups or ANOVA using Bonferroni *post hoc* correction for multiple groups. Data are expressed as the average $\pm$ SD.

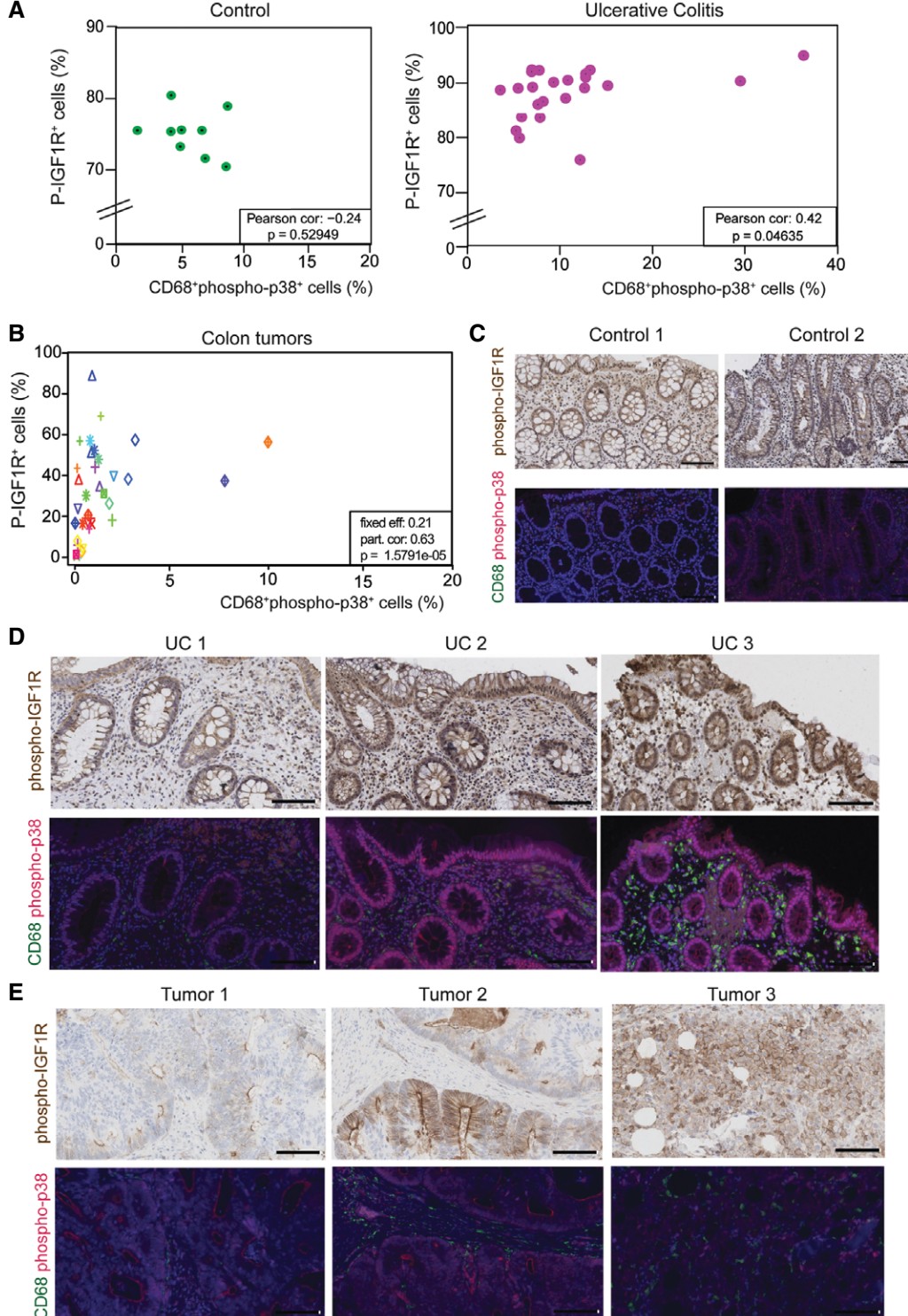

**Figure 8.   Activation of the IGF-1 pathway positively correlates with myeloid cells with active p38.**

A   Correlation between phospho-IGF1R+ cells and CD68+phospho-p38+ cells in colon tissues obtained from nine control individuals and 23 patients with ulcerative colitis. Statistical analysis was performed using the Pearson correlation test.

B   Correlation between phospho-IGF1R+ cells and CD68+phospho-p38+ cells in 48 tumors obtained from 25 patients with colon cancer. Same shape and color indicates tumors that are from the same patient. Statistical analysis was performed using the lmerTest.

C   Representative examples of colon sections from two healthy individuals stained for phospho-IGF1R and CD68/phospho-p38. Scale bars, 100 μm.

D   Representative examples of colon sections from three ulcerative colitis patients stained for phospho-IGF1R and CD68/phospho-p38. Scale bars, 100 μm.

E   Representative examples of sections from three colon adenocarcinomas stained for phospho-IGF1R and CD68/phospho-p38. Scale bars, 100 μm.

et al, 2011) and to induce macrophage migration in transwell chambers (Furundzija et al, 2010). On the other hand in the repair phase, no differences in epithelial damage could be observed between DSS-treated WT and IGF-1-$\Delta^{MC}$ mice. Notably, at this stage, inflammatory cell recruitment is not as prominent as in the acute pro-inflammatory phase, due to the withdrawal of DSS and the onset of repair mechanisms. We therefore hypothesize that IGF-1 contributes to a more efficient repair in WT mice, resulting in similar levels of epithelial damage compared with IGF-1-$\Delta^{MC}$ mice. We conclude that p38α-$\Delta^{MC}$ and IGF-1-$\Delta^{MC}$ mice do not phenocopy each other probably due to both the regulation by p38α of other inflammatory mediators, such as IL-1β, TNF-α, and IL-17, and the repair function of IGF-1.

The IGF-1 pathway has been linked to cancer and IBD by modulating the immune system as well as through its multi-functional involvement in the tumor microenvironment (Smith et al, 2011; Sanchez-Lopez et al, 2016). Aberrant IGF-1 signaling has been detected in various human cancer types, and there is evidence implicating IGF-1 receptor activity in cancer cell proliferation, migration, and invasion as well as in resistance to therapeutic agents (Pollak et al, 2004; Mourkioti & Rosenthal, 2005; Clayton et al, 2011; Vigneri et al, 2015; Spadaro et al, 2017). Moreover, a chemical compound that inhibits both IGF-1 and STAT3 signaling has been reported to reduce intestinal tumor burden induced by Apc deletion by affecting several stromal cells (Sanchez-Lopez et al, 2016). The impaired STAT3 signaling observed in p38α-$\Delta^{MC}$ and IGF-1-$\Delta^{MC}$ mice might be an indirect effect due to the reduced amount of STAT3-activating cytokines secreted by infiltrating immune cells, given the role of IGF-1 as a chemokine that increases immune cell recruitment to the colon. However, several reports have proposed that STAT3 phosphorylation can be induced by IGF-1 in various cell types including colonocytes (Zong et al, 2000; Zhang et al, 2006; Flashner-Abramson et al, 2016; Sanchez-Lopez et al, 2016; Yao et al, 2016; Xu et al, 2017). Consistent with the above, our results demonstrate that myeloid cell-specific downregulation of IGF-1 suffices to decrease colon tumorigenesis induced by AOM/DSS treatment.

Increased leukocyte infiltration is a hallmark of IBD and experimental colitis models, contributing to disease initiation and tissue damage (Abraham & Cho, 2009). Accordingly, the decreased inflammation observed in DSS-treated p38α-$\Delta^{MC}$ mice correlated with reduced colon infiltration of leukocytes. Ly6C$^{hi}$CCR2$^+$ monocytes are continuously generated in the bone marrow from hematopoietic stem cells and recruited to healthy and injured tissues, where they give rise to effector cells (Bain & Mowat, 2014). This monocyte population is reduced in p38α-$\Delta^{MC}$ mice compared to WT mice, even without treatment. This is likely to contribute to the observed phenotype, given the key role of inflammatory monocytes in triggering the recruitment of other immune cells as well as in the initiation of the adaptive immune response. The mobilization of Ly6C$^{hi}$CCR2$^+$ monocytes from the bone marrow depends on the CCL2-CCR2 chemokine axis, and deletion of either of these molecules markedly reduces circulating monocytes and ameliorates intestinal inflammation (Mowat & Bain, 2011; Bain & Mowat, 2014). We hypothesize that the differences in Ly6C$^{hi}$CCR2$^+$ monocytes observed in untreated p38α-$\Delta^{MC}$ mice are probably due to environmental factor exposure, such as microbiota (Bain & Mowat, 2014), leading to a state of controlled "physiological inflammation", which

is boosted upon DSS-induced inflammation and is linked to the differential expression of chemokines crucial for immune cell recruitment. Our results indicate that myeloid cell-specific downregulation of p38α results in a general downregulation of chemokines and cytokines in the colon. This could reflect the ability of myeloid p38α to regulate the production of these mediators, but it could also be due to different numbers of immune or epithelial cells present in p38α-$\Delta^{MC}$ mouse colons. Previous work has implicated p38α signaling in the regulation of chemokines by T helper lymphocytes and myeloid cells (Granata et al, 2006; Wong et al, 2007; Kim et al, 2008; Sun et al, 2008).

In conclusion, our results indicate that p38α signaling in myeloid cells supports colon inflammation and tumorigenesis, and identify myeloid IGF-1 as an important contributor to these processes. It seems likely that IGF-1 acts on various cell types, given that it is secreted to the microenvironment and that the IGF-1 receptor is ubiquitously expressed in normal tissues (Sanchez-Lopez et al, 2016). Of note, the different functions of p38α signaling in intestinal epithelial cells (Gupta et al, 2014) and in myeloid cells (this work) in the context of inflammation-associated colon tumorigenesis highlight the importance of selectively targeting specific cell types for therapeutic interventions in this pathway. Perhaps this has contributed to the disappointing results of p38α inhibitors in several clinical trials (Arthur & Ley, 2013; Patterson et al, 2014). Given the inherent difficulty to targeting a specific cell type, it is worth considering the alternative of targeting a combination of effector pathways with compounds that are already in clinical trials, often as separate medications. It has been proposed that some IBD patients could benefit from IGF-1 treatment due to its involvement in growth and repair processes, as well as anti-inflammatory effects (Zatorski et al, 2016). Taking into account the correlation that we observed in ulcerative colitis and colon tumor samples between myeloid cell recruitment and activation of the IGF-1 pathway, our results suggest that inflammatory features and local IGF-1 levels assessed from colonic biopsies should be taken into consideration. We propose that IGF-1 signaling might be an attractive target in the context of intestinal diseases with prominent inflammatory cell recruitment.

## Materials and Methods

The experiments conformed to the principles set out in the WMA Declaration of Helsinki and the Department of Health and Human Services Belmont Report.

### Human samples

Human CRC samples were obtained from the CNIO Biobank, and human UC samples were obtained and processed at the University Guadalajara Hospital (Brandt et al, 2018). Human samples were approved by the appropriate ethics committee, and informed consent was obtained from all subjects.

### Mice

p38α-$\Delta^{MC}$ mice were generated by crossing p38α$^{lox/lox}$ mice (Ventura et al, 2007) with LysM-Cre mice (Clausen et al, 1999). IGF-1-$\Delta^{MC}$

mice, carrying IGF-1$^{lox/lox}$ and LysM-Cre alleles, were generously provided by Nadia Rosenthal and Lina Wang (Tonkin *et al*, 2015). Eight- to 10-week-old mice of both genders were used in all experiments. Experimental groups were age- and sex-matched, and mice were randomly allocated to the treatment groups and killed by cervical dislocation. The mice used were in C57BL/6 background and housed according to national and European Union regulations in conventional housing conditions. Animal experiments were approved by the Animal Research Committee from the University of Barcelona.

### Induction of colitis and CAC in mice

For induction of acute colitis, mice received 1.5% DSS (molecular weight 36–50 kDa; MP Biomedicals #160110) *ad libitum* in the drinking water (Wirtz *et al*, 2007) for 6 days. At day 6, DSS was removed and mice were provided with regular drinking water. Body weight and DAI were recorded daily. DAI was determined by combining scores of weight loss, stool consistency, and rectal bleeding (divided by 3). Each score was determined as follows: change in weight (0: < 1%, 1: 1–5%, 2: 5–10%, 3: 10-15%, 4: > 15%), stool blood (0: absence, 2: presence, 4: gross bleeding), and stool consistency (0: formed and hard, 1: formed but soft, 2: loose stools, 3: mid diarrhea, watery, 4: diarrhea) as previously described (Cooper *et al*, 1993).

To study CAC, we applied a combination of the carcinogen AOM with repeated administration of DSS in the drinking water as previously described (Neufert *et al*, 2007). Mice received a single intraperitoneal injection of AOM (10 mg/kg; Sigma #A2853), and 5 days later, 1.5% DSS were administered for 5 days, followed by 14 days of regular drinking water. The DSS treatment was repeated for two additional cycles. Animals were sacrificed at the indicated time points, and colons were removed, separated from the cecum at the ileocecal junction, and flushed with cold PBS (Sigma #D1408) to remove feces and blood. After removing excess fat, the colons were opened longitudinally and samples for RNA and protein extracts were obtained by cutting a tiny piece horizontally beginning from the proximal up to the distal colon with a thickness of around 3 mm using sterile surgical blades (Quirumed #153-HB22). These samples were immediately shock-frozen in liquid nitrogen upon collection and stored at −80°C until use. The remaining colon (still the major part of the colon) was fixed as "swiss-rolls" in 10% formalin solution (Sigma #HT-501128) at RT overnight prior to paraffin embedding. In the case of mice killed after application of the tumorigenesis protocol, tumor sizes were measured using a digital caliper in a blinded manner, prior to sample collection (from colon and tumors) and fixation.

### Treatment of mice with recombinant IGF-1 and PQ401

The IGF1R inhibitor PQ401 was obtained from MedChem Express (#HY-13686) and was administered by oral gavage at a dose of 30 mg/kg in 30% Polyethylene glycol 400 (PEG400; Sigma #81170) with 0.5% Tween-80 (Sigma #P4780) and 0.5% Propylene glycol (Sigma #W294004). Control mice were treated in the same manner with the vehicle. In the case of DSS-induced acute colitis, the first dose was given 1 day prior to the start of DSS administration and was continued daily until mice were sacrificed. Mice treated with AOM/DSS to induce tumorigenesis were administered daily with PQ401 for 20 consecutive days. Oral gavage was performed as

described (Bertola *et al*, 2013). The dosing protocols were well tolerated by the animals.

For the treatment with IGF-1 during DSS-induced acute colitis, mice were administered daily with recombinant human IGF-1 (2 mg/kg; Peprotech #100-11) starting 1 day prior to the DSS administration and until mice were sacrificed. Control animals were injected with PBS.

### Isolation of peritoneal macrophages

Mice were sacrificed, and peritoneal cells were collected to assess the efficiency of p38α deletion. Prior to colon dissection, a small incision was made in the abdominal skin of the mice using surgical scissors. The skin was carefully retracted manually to expose the intact peritoneal wall, and 5 ml of cold PBS was injected through the peritoneal wall using 5-ml syringes (Sudelab #15922500) and 21G needles (BD Microlance #304432). After a gentle peritoneal massage to wash the peritoneal cavity, the PBS containing peritoneal cells were recovered with the syringe and cells were pelleted at 200 *g* for 5 min. For isolation and amplification of peritoneal macrophages, the cells were resuspended in 3 ml of Dulbecco's modified Eagle's medium (DMEM, Sigma #D5796) containing 1% penicillin–streptomycin (LabClinics #P11-010) and seeded onto non-treated tissue culture plates (Nunc #150239). After 30 min, mainly macrophages and monocytes adhere to the plates, which were then washed twice with ice-cold PBS to remove all dead and non-adherent cells, and complete DMEM was added containing 20% FBS (Thermo Scientific #E6541L) and 30% L-cell-conditioned medium derived from L-929 mouse fibroblasts (ATCC) as a source of M-CSF and 1% penicillin–streptomycin (LabClinics #P11-010). Cells were left to proliferate in complete DMEM for 1 or 2 days, depending on the number of macrophages on the plates. Finally, the plates were washed using ice-cold PBS and collected with lysis buffer on ice using a cell scraper (Costar #3008), as described below in the "Protein extraction and immunoblotting" section.

### Isolation of macrophages from colonic lamina propria

To obtain colon macrophages, the large intestine was excised, washed in PBS, and opened longitudinally. Colons were washed again in Hank's balanced salt solution (HBSS; Gibco #14175-137) containing 2% FBS, and cut into approximately 0.5-cm sections using sterile surgical blades. These pieces were then shaken vigorously in 10 ml HBSS with 2% FBS and the supernatant was discarded. HBSS containing 2 mM EDTA was then added, and the tube was placed in a shaking incubator for 15 min at 37°C, before being shaken vigorously and the supernatant discarded. Prior to a second incubation with HBSS/EDTA at 37°C for 30 min, the tissue was washed once with HBSS, shaken vigorously and the supernatant discarded. After this second incubation, the step was repeated and the remaining tissue was digested in pre-warmed RPMI 1640 (Sigma #R8758) containing 2 mM L-glutamine (LabClinics #x0550), 100 μg/ml penicillin, 100 μg/ml streptomycin, and 10% FBS (complete RPMI 1640 medium) supplemented with 1.25 mg/ml collagenase D (Roche # 11088866001), 0.85 mg/ml collagenase V (Sigma #C9263), 1 mg/ml dispase (Gibco #17105-041), and 30 μg/ml DNase (Roche #10104159001) for 30–45 min in a shaking incubator at 37°C, shaken vigorously every 7 min until

no intact tissue was left, but without exceeding 45 min of total incubation time in the presence of the enzyme mix. The resulting cell suspension was passed through a 40-μm cell strainer (BD Falcon # 352340) and then washed twice in complete RPMI 1640 (centrifuged 5 min at 200 $g$ and resuspended in complete RPMI 1640). This cell suspension was seeded onto non-treated plastic tissue culture plates in order to select and enrich monocyte/macrophage population from the whole lamina propria. The cells were resuspended in complete RPMI 1640 medium and were pelleted again by centrifugation and recovered in DMEM containing 1% penicillin–streptomycin. After 30 min, the cells were collected and used for purification of total RNA.

## Generation and treatment of BMDMs

Bone marrow precursor cells were isolated from femurs and tibias of mice and cultured in complete Dulbecco's modified Eagle's medium (DMEM) as previously described (Bailon et al, 2010). For analysis of IGF-1 protein at basal conditions, medium was renewed after 5 days of differentiation and supernatants and cells were collected at day 7 after bone marrow extraction. For analysis in the presence of inhibitors or after stimulation, cells were deprived of macrophage colony-stimulating factor (M-CSF) for 18 h in DMEM (10% FBS, 1% penicillin–streptomycin) at day 6 of differentiation to synchronize the culture and render the cells quiescent. The inhibitors PH797804 (2 μM; Selleckchem #S2726), SB203580 (10 μM; Axon Medchem #AX1363), and LY2228820 (0.1 μM; Axon Medchem #1895) were added 1 h before stimulation with lipopolysaccharide (LPS) (10 ng/ml; Sigma #L4005) or interleukin (IL)-4 (10 ng/ml; R&D Systems).

## Protein extraction and immunoblotting

Peritoneal macrophages and BMDMs were collected in lysis buffer containing 1% NP-40, 150 mM NaCl, 50 mM Tris–HCl pH 7.5, 2 mM EDTA, 2 mM EGTA, 20 mM sodium fluoride, 2 mM PMSF, 2 mM sodium orthovanadate, 1 mM DTT, and protease inhibitor cocktail (1:100; Sigma #P8340). Frozen colon samples collected as described above were defrosted at 4°C and lysed in the lysis buffer using the Precellys instrument (Bertin Technologies #03119.200.RD000). Cell and tissue lysates were left for 15 min on a rotatory mixer at 4°C, prior to centrifugation at 20,000 $g$ for 10 min at 4°C. Supernatants were collected and quantified using Bradford reagent (Sigma #B6916) with BSA as standard. Samples were then either directly used for immunoblotting or enzyme-linked immunosorbent assay (ELISA) or stored at −80°C until use.

For immunoblotting, 40 μg of lysates was boiled for 5 min at 95°C and separated by SDS–PAGE prior to transfer onto a nitrocellulose membrane (Whatman #10401396). Membranes were blocked with 5% non-fat milk in TBST buffer (50 mM Tris–HCl pH 7.5, 150 mM NaCl, and 0.1% Tween-20). After three washes with TBST, membranes were incubated with the primary antibodies overnight at 4°C in TBST containing 5% of BSA. The following antibodies were used: p38α (1:1,000; Cell Signaling #9218) and GAPDH (1:5,000, Sigma #SAB2100894) as loading control. After three washes with PBST, membranes were incubated with Alexa Fluor 680 or 800-conjugated antibodies (Invitrogen; 1:5,000) for 1 h at

RT prior to visualization using Odyssey Infrared Imaging System (Li-Cor, Biosciences).

## ELISA and chemokine array analysis

For detection of IGF-1, supernatants derived from BMDMs were collected and centrifuged at 7,500 $g$ for 5 min at 4°C to remove dead cells and cellular debris, and diluted 1:5 for analysis using the mouse/rat IGF-1 Quantikine ELISA kit following the manufacturer's instructions (R&D Systems #MG100). The adhered BMDMs were washed once with ice-cold PBS and directly lysed on the plates using a cell scraper on ice. Protein extracts from BMDMs and colon lysates were analyzed by ELISA using 100 μg per sample and well.

IL-1β and TNF-α were analyzed by using 100 μg per sample and well and the mouse IL-1β/IL-1F2 Quantikine ELISA kit (R&D Systems #MLB00C) or mouse TNF-α Platinum ELISA kit (eBioscience #BMS607) following the manufacturer's instructions.

For the chemokine array, whole colon extracts were obtained from mice either WT or deficient for p38α in myeloid cells (five different mice each), and a total of 400 μg of protein was pooled (80 μg protein per mouse) and analyzed following the manufacturer's instructions (R&D Systems #ARY020).

For the cytokine array, whole colon or tumor extracts were obtained from mice either WT or deficient for p38α in myeloid cells (eight different mice each, four male and four female per genotype), and a total of 200 μg of protein was pooled for each membrane (25 μg protein per mouse) and analyzed following the manufacturer's instructions (R&D Systems #ARY006).

## Determination of IGF-1 in blood serum

Blood was collected by cardiac puncture, and serum samples were obtained using SST tubes (BD #365968). IGF-1 protein concentration in serum was determined using the mouse/rat IGF-1 Quantikine ELISA kit following the manufacturer's instructions (R&D Systems #MG100).

## Immunohistochemistry and immunofluorescence

Formalin-fixed and paraffin-embedded mouse colon sections were stained with antibodies against CD45 (BD Biosciences, #550539; 1:100, overnight at 4°C), F4/80 (eBiosciences #14-4801; 1:50, 2 h at RT), CD3 (Dako #A0452; 1:10, 120 min at RT), MPO (Dako #A0398; 1:1,000, 30 min at RT), F4/80 (eBiosciences #14-4801; 1:50, 2 h at RT), STAT3 (phospho Tyr705; Cell Signaling #9145; 1:200, 90 min at RT), IGF1R (phospho Y1161; Abcam #ab39398; 1:500, 60 min at RT), cleaved caspase-3 (#9661; 1:200, 1 h RT), and Ki67 (Novocastra #NCL-Ki67p; 1:500, 1 h at RT). The secondary antibodies used were as follows: HRP-conjugated anti-rabbit (ImmunoLogic #DPVR110HRP, 45 min at RT), anti-mouse (Dako #P0447; 1:100, 30 min at RT), anti-rat (Dako #P0450; 1:75, 30 min at RT), and anti-goat (Dako #P0449; 1:80, 30 min at RT). To stain bone marrow cells, formalin-fixed bones were first decalcificated for 2 weeks using Osteosoft (Millipore #101728). Human tissues were stained with phospho-Y1161 IGF1R (Abcam #ab39398; 1:300, overnight at 4°C) and HRP-conjugated anti-rabbit (ImmunoLogic #DPVR110HRP, 45 min at RT). For the immunofluorescence

double staining of CD68 and phospho-p38, the phospho-p38 antibody (Cell Signaling #4631) was added 1:100 to the ready-to-use CD68 antibody (Dako #IS613) and incubated overnight at 4°C. Slides were then incubated with anti-mouse Alexa Fluor 488 and anti-rabbit Alexa Fluor 588 secondary antibodies (Invitrogen; 1:400) for 1 h at RT.

Slides were scanned using the digital scanner Nanozoomer 2.0HT (Hamamatsu) with a 40× objective. The number of positively stained cells was calculated from several high-magnification fields per animal. For the analysis of the epithelium of untreated or DSS-treated mice, most of the distal part of the epithelium of each animal was analyzed. For the analysis of the non-tumoral epithelium in AOM/DSS-treated mice, most of the distal part of the epithelium was analyzed, excluding the tumors. Of note, in epithelium from both treated and untreated mice, the regions analyzed were limited to the mucosal layer, excluding the muscular and serosa layers. In order to compare tumors of the same size, several tumors of 3 mm of size were cut out of the distal part of the colons after sacrifice of the animals and embedded in paraffin separately. High-magnification images were taken for the whole tumor tissue stained per slide and animal analyzed.

The regions of interest were exported using ImageJ software, and detection of positive staining and cell number was performed with ImageJ software using the color deconvolution plug-in that has a built-in vector for separating hematoxylin (H) and diaminobenzidine (DAB) staining. After color deconvolution, DAB images are processed separately. Suitable threshold levels of DAB were determined for each staining and kept constant for all analysis.

The quantification of phospho-IGF1R-positive cells for different staining intensities was performed with TMarker (Schuffler *et al*, 2013) using the plug-in for "color deconvolution" and "cancer nucleus classification". Several training images were used in order to TMarker classification of the samples using its active learning algorithm prior to performing the analysis of the samples from several high-magnification fields.

For analysis of apoptotic cells by TUNEL assays, paraffin-embedded samples were processed using the *In Situ* Cell Death Detection Kit (Roche #11684795910) according to the manufacturer's instructions.

To confirm the purity of macrophages isolated from lamina propria, cells were left to adhere to the plastic petri dishes for 30 min and were collected and seeded onto 8-well glass chamber slides (Millipore #PEZGS0816) for immunofluorescence staining. Then, cells were left to adhere for 30 min, the culture media were removed, and cells were immediately fixed with ice-cold methanol (100%) for 5 min at RT. Methanol was removed, and cells were washed three times with PBS and incubated with 100 μl of blocking buffer [PBS with 3% BSA, 0.1% Triton X-100 (Sigma #T8787), and 1:50 dilution of CD16/32 blocking antibody (eBioscience #16-0161-85)] with shaking for 30 min. The CD16/32-containing blocking buffer was removed and replaced with 100 μl of blocking buffer without CD16/32 but containing the primary antibody. After incubation for 1 h with shaking at RT, cells were washed three times with blocking buffer. A secondary antibody diluted in blocking buffer was then added for 1 h, except in the case of conjugated antibodies. After finishing the incubations with the antibodies required, cells were washed three times in blocking buffer prior to

mounting them on coverslips using ProLong Gold antifade reagent with DAPI (Invitrogen #P36935). The primary antibodies used were F4/80 (1:100; eBioscience #123113) and CD115 (1:100; BioLegend #135520), and the secondary antibodies used were Alexa Fluor 488 (1:400; Invitrogen #A21441) and Alexa Fluor 555 (1:400; Invitrogen #A21434).

## Epithelial damage scoring

H&E-stained slides were scanned using the digital scanner Nanozoomer 2.0HT (Hamamatsu) with a 40× objective. Epithelial damage and inflammation were determined using a previously described scoring system (Gupta *et al*, 2014). Briefly, the epithelial damage observed in the colon was first scored (1–4) using the following criteria: (1) intact crypts, (2) basal one-third damaged, (3) basal two-thirds damaged, and (4) damaged surface epithelium. The colon sample was given the score of the highest epithelial damage observed in the entire intestinal mucosa. Then, the extent of epithelial damage as percentage of the whole colon was assessed and scored (1–4) as follows: (1) 1–25%, (2) 26–50%, (3) 51–75%, and (4) 76–100%. The two scores were added up and divided by two, resulting in the final epithelial damage score.

## Preparation of bone marrow and colon for flow cytometry analysis

Bone marrow cells were isolated as previously described (Bailon *et al*, 2010). After collection of the cells out of the bones, cells were transferred to 15-ml Falcon tubes, centrifuged (200 *g*, 5 min), resuspended in 5 ml red cell lysis (RCL) buffer (150 mM ammonium chloride, 1 mM potassium bicarbonate, 0.1 mM EDTA in distilled water; pH = 7.2–7.4), and incubated for 5 min on RT. Ice-cold DMEM containing 10% FBS was added, and cells were centrifuged once more (200 *g*, 5 min) and resuspended in ice-cold fluorescence-activated cell sorting (FACS) buffer, followed by staining.

For the analysis of intestinal myeloid cells, the colon was processed as described in "Isolation of macrophages from colonic lamina propria", but instead of leaving the cell suspension to adhere on plates, the cells were centrifuged (200 *g*, 5 min), washed once in ice-cold FACS buffer, and resuspended in the same buffer, followed by staining.

After blocking the Fc receptors of the cells for 15 min at 4°C in FACS buffer containing the CD16/CD32 antibody (1:100, eBioscience #16-0161-85), the cells were stained with combinations of fluorescence-labeled antibodies to the cell surface markers for 30 min (4°C): CD45 (1:400; eBioscience #47-0451-80), CD11b (1:200; BioLegend #101241), Ly6C (1:400; BD Pharmingen #560595), and CCR2 (1:100; R&D Systems #FAB5538F-025), as well as Lineage-Cocktail (1:100; BioLegend #88-77772-72), Sca-1 (1:100; BioLegend #108114), c-Kit (1:100; BioLegend #108114), CD34 (1:100; BioLegend #11-0341-81), and CD127 (1:100; BioLegend #12-1271-81). Cells were then stained for viability using the LIVE/DEAD Fixable Yellow Dead Cell Stain Kit (Life Technologies #L-34959) or DAPI (Invitrogen #MP-36930) following the manufacturer's instructions and analyzed using Gallios (Beckman Coulter) or Cytoflex (Beckman Coulter) flow cytometers or sorted by FACS Aria 2.0 (BD Biosciences). All experiments were

analyzed using FlowJo software (Phoenix Flow Systems, Inc., San Diego).

### RNA extraction and gene expression analysis

Total RNA was extracted from BMDMs using PureLink RNA mini kit (Ambion #12183018A) and on-column DNase treatment (Ambion #12185-010) following the manufacturer's instructions. Colon samples were collected as described above and defrosted at 4°C. Then, extraction was performed using RNeasy mini kit (Qiagen #74104) and DNase I treatment (Roche #04716728001) following user's manual instructions. Samples were quantified by NanoDrop, and reverse transcription was performed using 1 μg of total RNA using Superscript II reverse transcriptase (Invitrogen #18064-014), RNasin (Promega #N2111), and random primers (Invitrogen #48190-011) following Invitrogen user's instructions for cDNA synthesis using SuperScript II RT. qRT–PCR was performed in triplicates using 4 μl of 1/16 diluted cDNA and SYBR Green (Life Technologies #4472954) in 10 μl final volume using Quant-Studio 6 Flex Instrument (Life Technologies).

For intestinal macrophages isolated from the colonic lamina propria as described above, total RNA was extracted using TRIzol (Ambion #15596018) and Pellet Paint Co-Precipitant (Millipore #69049). TRIzol (300 μl) was added to each well containing intestinal macrophages, incubated for 5 min at RT, and chloroform (60 μl) was added. The samples were shaken vigorously for 15 s and allowed to stand for 5 min at RT prior to centrifugation for 15 min at 12,000 *g* (4°C). The colorless aqueous upper phase containing the RNA was then transferred to a new tube. Pellet Paint (2 μl) and 1 volume of isopropanol (Sigma #I9516) were added to the samples and incubated on ice for 5 min. Samples were then centrifuged for 30 min at 12,000 *g* (4°C). Supernatants were removed, and the pellets were washed with 300 μl of ethanol by vortexing. The tube was put 180° relative to the previous step and centrifuged for 10 min at 12,000 *g*. The supernatant was removed, and 15 μl of RNase-free water was added to the pellet and left for 5 min at RT. Samples were vortexed for 1 min and spun down briefly. Samples were pipetted up and down to further homogenize and spun down again. DNase I treatment was performed following user's manual instructions (Roche #04716728001). Samples were quantified by NanoDrop and reverse-transcribed using 1 μg of total RNA, RNAsin, random primers, and Superscript IV reverse transcriptase (Invitrogen #18090050) as recommended by Invitrogen. qRT–PCR was performed in triplicates using 4 μl of 1/32 diluted cDNA and SYBR Green in 10 μl final volume using Quant-Studio 6 Flex Instrument. Data were obtained as relative mRNA levels normalized in each sample to the GAPDH expression level. The primers used are listed in Appendix Table S1.

### Analysis of p38α deletion

Genomic DNA was isolated using standard phenol–chloroform extraction from bone marrow-sorted cells. RNA was isolated from intestinal macrophages as described in "RNA extraction and gene expression analysis" above. PCR analysis was performed with primers specific for exon 2 (floxed) and exon 12 (as a control) of the *Mapk14* gene encoding p38α. Relative

**The paper explained**

**Problem**
It is estimated that each year more than one million people will develop colorectal cancer worldwide and the mortality of this disease is approximately 35% in the developing world. Although most cases occur sporadically, chronic inflammation is one of the main reasons for developing neoplasia in the colon. Colitis-associated cancer is a type of colorectal cancer induced by inflammatory disorders, such as inflammatory bowel disease. The causes and molecular mechanisms of the pathogenesis of this disease are complex and heterogeneous. Differential diagnosis of patients is of pivotal importance for a personalized clinical management, as each case involves specific therapeutic strategies due to the heterogeneity of this disease.

**Results**
Our results demonstrate that myeloid cells rely on p38α signaling to promote inflammation-associated colon tumorigenesis. We show that myeloid p38α is critical for the local production of chemo-attractants, which in turn are required for immune cell recruitment to the colon. We also identify IGF-1 as a novel mediator of p38α signaling in myeloid cells contributing to intestinal inflammation. Moreover, genetic or pharmacological inhibition of IGF-1 suppresses inflammatory cell recruitment and reduces colitis-associated tumor burden. Importantly, we observed a correlation between IGF-1 pathway activation and the infiltration of macrophages with active p38 in samples from ulcerative colitis and colon cancer patients.

**Impact**
Our results highlight the potential therapeutic interest of inhibiting the p38α pathway in myeloid cells, especially in tumors associated with chronic inflammation, given that macrophages are prominently found in and recruited to tumors. We therefore propose that targeting extracellular effectors of p38α signaling in myeloid cells, which are implicated in disease pathogenesis such as IGF-1, might overcome the difficulties associated with cell type-specific targeting. We suggest that therapeutic decisions should consider the inflammatory features and local IGF-1 levels assessed in colonic biopsies from patients suffering from inflammatory bowel disease or colitis-associated cancer.

amount of exon 2 versus exon 12 was determined. Primers are listed in the Appendix Table S1.

### Statistics

Data are expressed as average ± SD. Statistical analysis was performed by using Mann–Whitney test for the comparison of two groups or ANOVA using Bonferroni *post hoc* correction for multiple groups using GraphPad Prism Software 6 (GraphPad Software, Inc., La Jolla, CA). *P*-values are indicated in the figures.

For the correlation analysis of samples from ulcerative colitis patients, Pearson's correlations were computed after applying the square root transformation to the raw percentages. The correlation coefficient and *P*-values were computed using the function cor.test from R (R Core Team, 2013).

For the correlation analysis of human colon tumors, we fitted several linear mixed-effects models by taking CD68[+]phospho-p38[+] variables as response and phospho-IGF1R variables as fixed effect. Confounding factors such as plate (determined by sample location on the array) and patient (determined by sample ID) were used as fixed effect and random effect, respectively. Confidence intervals

and *P*-values were obtained using the R package lmerTest (Kuznetsova *et al*, 2017) by applying a likelihood ratio test. Pearson's partial correlation coefficients, adjusting by plate, were also computed. The square root transformation of both variables was considered before fitting the models.

**Expanded View** for this article is available online.

## Acknowledgements
We are grateful to Nadia Rosenthal and Lina Wang for providing IGF-1-Δ$^{MC}$ mice. C.Y. is very grateful to Joan Guinovart for his personal and financial support. We acknowledge the excellent technical assistance of Neus Prats and the IRB Histology facility members, Camille Stephan-Otto Attolini and Adria Caballe from the IRB Biostatistics/Bioinformatics facility, and Jaume Comas and the UB fluorescence-activated cell sorting facility. We thank Teresa Rodrigo Calduch, head of the Unitat d'Experimentació Animal de Farmàcia de la Universitat de Barcelona. We are grateful to the Nebreda group at IRB Barcelona for their support and many useful discussions. This work was supported by grants from the European Commission (Advanced ERC 294665), Fundación Marató TV3 (20133430), Spanish MINECO (SAF2016-81043-R), and AGAUR (2014 SRG-535 and 2017 SGR-557). C.Y. acknowledges a "La Caixa" predoctoral fellowship. IRB Barcelona is the recipient of institutional funding from MINECO (Government of Spain) through the Centres of Excellence Severo Ochoa award and from the CERCA Program of the Catalan Government.

## Author contributions
CY and ARN designed the project with input from MC. CY performed most experiments. MC-R performed some experiments for the manuscript revision. EL performed IHC analysis. ND and CP provided human samples and advice to analyze them. CY and ARN analyzed the data and wrote the manuscript with contributions from MC and other authors. ARN provided funding and supervised the study.

## Conflict of interest
The authors declare that they have no conflict of interest.

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
