## [Review Process File · EMBO Molecular Medicine]

Myeloid p38 α signaling promotes intestinal IGF-1 production and inflammation-associated tumorigenesis

Catrin Youssif, Monica Cubillos-Rojas, Mònica Comalada, Elisabeth Llonch, Cristian Perna, Nabil Djouder and Angel R. Nebreda

Review timeline:

Submission date:	19 August 2017
Editorial Decision:	21 September 2017
Revision received:	01 April 2018
Editorial Decision:	07 May 2018
Revision received:	16 May 2018
Accepted:	18 May 2018

Editor: Céline Carret

Transaction Report:

1st Editorial Decision

21 September 2017

Thank you for the submission of your manuscript to EMBO Molecular Medicine. We have now heard back from the two referees whom we asked to evaluate your manuscript.

As you will see from the reports below, the referees find the topic of your study of potential interest. However, they do raise substantial concerns on your work, which should be convincingly addressed in a major revision of the present manuscript. Of particular importance, we would like to draw your attention on the overlapping concerns regarding the limited mechanistic details that must be strengthened: the link between p38a and IGF1, the effects on epithelium / colitis and the inflammatory cells recruitment. Besides, both referees agree that more patient data would greatly increase the clinical relevance of the study, and while ref. 2 suggests using organoids, this referee agreed upon our cross-commenting exercise with ref. 1 that if better *in vivo* mechanisms would be provided, organoids would not be needed.

We would welcome the submission of a revised version within three months for further consideration and would like to encourage you to address all the criticisms raised as suggested to improve conclusiveness and clarity. However, we do realize that addressing all these issues would require a lot of additional work and experimentation and I am unsure whether you will be able or willing to address those and return a revised manuscript within the 3 months deadline.

Please note that EMBO Molecular Medicine strongly supports a single round of revision and that, as acceptance or rejection of the manuscript will depend on another round of review, your responses should be as complete as possible.

I would understand your decision if you chose to rather seek rapid publication elsewhere at this stage. If this turns out to be the case, I would appreciate an email to this effect. Otherwise, I look forward to receiving your revised manuscript.

***** Reviewer's comments *****

Referee #1 (Comments on Novelty/Model System for Author):

- 1) Experiments well performed and controlled
- 2) There is a manuscript describing myeloid-specific deletion of p38, which has however not described IGF1 as a target.
- 3) No patient data are shown; however important results for the use of p38 inhibitors in the clinic.

Referee #1 (Remarks for Author):

Youssif et al. present interesting data on the role of p38 in myeloid cells. Deletion of p38 in myeloid cells reduced damage after DSS-induced colitis and AOM/DSS-induced tumor growth. This was accompanied by reduced chemokine production and reduced numbers of macrophages in the inflamed colon and tumors. The authors were able to show that IGF1 is reduced in macrophages of p38dMC mice and myeloid-specific deletion of IGF1 resulted in similar phenotypes as p38dMC mice, suggesting that IGF1 is downstream of p38 in myeloid cells. Although some of the *in vivo* phenotypes are comparable, there remain differences especially in the DSS model and some questions remain to be answered.

- 1) There is strong circumstantial evidence that IGF1 might be downstream of p38. However the direct mechanistic involvement is weak and based mostly on inhibitor studies. To strengthen this findings the authors should treat p38dMC mice with IGF1 and demonstrate that the observed defects can be rescued. Moreover, as pointed out by the authors, there are many cells capable of producing IGF1 and, as shown by IGFR staining, many target cells in the colon. To be able to delineate the contribution of autocrine vs paracrine IGF1 and explore synergistic effects of p38 and IGF1 inhibition, the authors should treat p38dMC mice with the IGFR inhibitor PQ401.
- 2) Another important aspect that needs clarification is the role of macrophage recruitment. The authors focus on a reduced number of CCR2⁺ Ly6C⁺ cells in the bone marrow that may effect on recruitment to the inflamed colon. The mechanism of this is however completely unclear. Are the levels of IGF1 changed systemically in p38dMC mice to induce effects on bone marrow? Is the effect on BM monocytes directly caused by IGF1? Moreover, is Lys-Cre active in Ly6Chi/CCR2⁺ cells?
- 3) The chemokines are already reduced in baseline colon and macrophages. However, the number of F4/80⁺ cells in healthy colon does not seem to be changed (Figs 1F, 2C). Can this be really the cause for the reduced migration of inflammatory monocytes to the colon? How are chemokine levels in early DSS treated mice? What about other important inflammatory mediators in DSS colitis, such as the Th17 inflammatory response? Otsuka et. al have previously reported reduced levels of important inflammatory cytokines (e.g. IL6, IL12, IL1b, TNF etc.) in p38dMC mice after DSS colitis. How does this fit into the overall picture proposed here? There is no mention in the proposed model or discussion.
- 4) The authors show that there are reduced numbers of wound-healing macrophages (and reduced IGF1) in p38dMC mice after DSS colitis. However, epithelial regeneration is not affected and seems to be even better compared to wt mice. How do the authors explain/discuss these findings?
- 5) Showing IGF1 / p38 stainings in CRC and/or colitis/IBD patients would have strengthen the medical relevance of the study.
- 6) How is phospho pSTAT3 quantified? It would make sense to delineate staining on epithelial vs stromal cells? Can pSTAT3 be directly induced by IGF1 in colonocytes or myeloid cells? How is pSTAT3 in IGF1 dMC mice?
- 7) Figure 1a and 4b look identical, as well as Suppl. Fig. 4E and F. Please clarify.

- 8) F4/80 staining in healthy colon is generally weak. Only 1 or 2 cells / section in healthy colon is very few. Did the staining work properly?
- 9) What does % of cells mean (e.g. Fig. 1E,F)? Are Serosa cells also included in counting? Should the sections rather be counted per colon area?
- 10) The following pictures are too small and higher magnifications should be shown: Fig. 1F, Fig. 2D-F
- 11) The reference to Suppl. Fig1A in the result and the respective Figure do not match.

Referee #2 (Comments on Novelty/Model System for Author):

Since the effect of myeloid p38 during DSS colitis has been published the novelty of the findings is not high. Although the contribution of IGF1 to this is new, this should be tested more rigorously in additional models system (organoids) and also alternative mechanisms should be more carefully evaluated.

Referee #2 (Remarks for Author):

In this manuscript, Youssif et al examine the role of myeloid p38a signaling in a model of colitis and colitis-associated tumorigenesis. Contrary to epithelial p38a signaling which is tumor suppressive during tumor initiation, now authors report that myeloid p38a signaling has a tumor promoting function. Accordingly, epithelial damage and inflammation was reduced in p38aDMC mice. Moreover, they suggest that p38a in myeloid cells controls IGF1 expression, which acts as chemo-attractant for inflammatory cells, and seems important for colitis-associated tumorigenesis. Blocking IGF1 signaling using chemical inhibitor reduced epithelial damage and tumor load. The p38a-IGF1 link during colon tumorigenesis is interesting but there are several concerns:

1. The phenotype that p38a depletion in myeloid cells reduces DSS induced epithelial damage and impairs number of inflammatory cells in the mucosa was already published, as author also cited that work. However, the link between p38a and IGF1 signaling in this context and its impact on tumorigenesis is new. Nevertheless, the suggested mechanism is not entirely convincing. If the authors propose that reduced myeloid cell recruitment is responsible for improved outcome of DSS colitis, this cannot entirely explain the phenotype as it has been shown that macrophages are required for proper wound healing. In particular Fig. 4C is not very convincing in this context. It is not clear if the proliferation rate shown is referring to epithelial cells. As the histology in the 7 day DSS panels does not show any epithelial cells but only a big ulcerations. Seeing this it also very surprising to note that at 13 days the mucosa in the knockout samples is completely healed whereas there is still massive inflammation in the wt mice. Are these really representative stainings? A lower magnification should be provided to see the entire distal colon of the mice analyzed. In addition to the clinical data (DAI) the authors should provide a rigorous histological assessment of the histological damage at both time points. Weight loss and tissue damage does not necessarily correlate between male and female mice. Female mice are more resistant to weight loss, yet they can show the same histological damage as male mice. If wound healing/regeneration is impaired in ko mice, initial tissue damage should be less otherwise mice would show more weight loss and not less.
2. In tumors no difference was observed in tumor size between Wt and p38a KO mice (Suppl Fig1D). However, at this point both phospho-IGF1R and Phospho-Stat3 seems to be more in WT tumors compared to p38aKO tumors. As authors wrote in the text, that Stat3 and IGF1 signaling both are important for cell proliferation and considering the reduced Stat3 and IGF1 signaling in KO tumors, one would expect to have less proliferation in tumors and perhaps also smaller tumors. Proliferation and apoptosis rates need to be determined in these tumors.
3. Differences in the cytokines production (IL1b and TNFa, Fig 2F) upon DSS treatment are very moderate. Moreover, it was shown that IL1b KO mice are more susceptible to DSS-induced colitis (PMID: 23793223), which is somewhat contradictory to what is observed in p38a KO mice. Perhaps these differences can be attributed to other cytokines and chemokines regulated by p38a. IL1b and TNFa are usually associated directly with NFkB signaling (also shown by PMID 20080092), rather than Stat3 signaling which could be regulated by these cytokines indirectly. Author should also examine IL6 family cytokines, which are direct inducers of Stat3 signaling.

4. Regulation of IGF1 by p38a in BMDMs is very clear as shown in Fig 3 but the differences in IGF1 levels in vivo during colitis are very mild. Similarly, P-IGF1R staining in the p38a KO tumors has slightly reduced expression compared to WT tumors. Authors should also measure IGF1 in the tumors.

5. In the entire manuscript, number of mice used in the experiments is mentioned but how many times experiments were repeated, is not mentioned.

6. Suppl Fig 4B and C: Effect of IGF1 depletion in myeloid cells has very little effect on body weight loss and DAI compared to p38a KO mice. 31 mice are used for DAI quantification, but still differences between groups are not impressive. Is this pool data or from one single experiment? If it's a pool, then authors should show individual experiments.

7. Compared to p38a KO mice, the differences in phospho-Stat3 staining in IGF1 KO mice does not seem significant. How is Stat3 signaling regulated in these experiments? Similarly, IGF1 KO mice do not have (or very little) any effect on P-IGF1R as shown in Fig S4.

8. IGF1 KO mice have reduced tumor load without affecting the size. In fig 6A, 22 mice were used. Is this pooled data? If not, shown individual experiments. Interestingly, F4/80+ cells were significantly reduced in tumors from IGF1 KO mice but was not in non-tumoral area of these mice despite having similar percentage of F4/80+ cells in tumors vs non-tumor. Discuss this tumor specific effect.

9. Mechanistic details how p38a-IGF1 signaling axis affect epithelial repair and tumorigenesis is not fully delineated. Recently it was shown that circulating IGF1 can control colonic stem cell function in diabetic enteropathy (PMID: 26431183). Authors should check number of stem cells during DSS-induced colitis, repair and tumors both in p38a and IGF1 KO mice. In addition, 3D organoids from colon should be used to functionally examine the role of IGF1 on stem cells as this modes can serve as a toll to examine regeneration.

10. In Fig1B, statistical data on rectal prolapse comparison between wt and p38a ko mice should be provided.

1st Revision - authors' response

01 April 2018

Reviewer #1

Youssif et al. present interesting data on the role of p38 in myeloid cells. Deletion of p38 in myeloid cells reduced damage after DSS-induced colitis and AOM/DSS-induced tumor growth. This was accompanied by reduced chemokine production and reduced numbers of macrophages in the inflamed colon and tumors. The authors were able to show that IGF1 is reduced in macrophages of p38dMC mice and myeloid-specific deletion of IGF1 resulted in similar phenotypes as p38dMC mice, suggesting that IGF1 is downstream of p38 in myeloid cells. Although some of the in vivo phenotypes are comparable, there remain differences especially in the DSS model and some questions remain to be answered.

1) There is strong circumstantial evidence that IGF1 might be downstream of p38. However the direct mechanistic involvement is weak and based mostly on inhibitor studies. To strengthen this findings the authors should treat p38dMC mice with IGF1 and demonstrate that the observed defects can be rescued. Moreover, as pointed out by the authors, there are many cells capable of producing IGF1 and, as shown by IGFR staining, many target cells in the colon. To be able to delineate the contribution of autocrine vs paracrine IGF1 and explore synergistic effects of p38 and IGF1 inhibition, the authors should treat p38dMC mice with the IGFR inhibitor PQ401.

We have now performed the two experiments. As illustrated in Figure 6, treatment of p38a-DMC mice with recombinant IGF-1 rescued several phenotypes induced by DSS in p38a-DMC mice, such as the disease activity index (Fig 6A), levels of pro-inflammatory monocytes in the bone marrow (Fig 6B), epithelial damage (Fig 6C) and STAT3 phosphorylation (Fig 6D). As suggested by the reviewer, we also treated p38a-DMC mice with the IGFR inhibitor PQ401 (Appendix Fig S4E-S4G). However, although the colon of p38a-DMC mice still show substantial activation of the IGFR

pathway, we could not observe any synergistic effect of p38a downregulation in macrophages with the systemic inhibition of the IGFR pathway. However, it should be noted that IGF-1 levels in the colon of DSS-treated p38a-DMC mice are already reduced to about the levels of the normal epithelium (Fig 4A). These observations together with the results observed in the IGF-1-DMC mice suggest that although several cell types can produce IGF-1, the IGF-1 produced by myeloid cells is likely to substantially contribute to the DSS-induced intestinal inflammation.

2) Another important aspect that needs clarification is the role of macrophage recruitment. The authors focus on a reduced number of CCR2+ Ly6C+ cells in the bone marrow that may effect on recruitment to the inflamed colon. The mechanism of this is however completely unclear. Are the levels of IGF1 changed systemically in p38dMC mice to induce effects on bone marrow? Is the effect on BM monocytes directly caused by IGF1? Moreover, is Lys-Cre active in Ly6Chi/CCR2+ cells?

We thank the reviewer for raising this issue, which we have addressed by performing further experiments. As expected, given that the primary source of systemic IGF-1 is the liver (Gow, Sester et al., 2010), we could not observe any differences in IGF-1 levels of serum from WT and p38a-DMC mice (Appendix Fig S3B). However, further analysis indicated that the IGF-1 pathway was less active in the bone marrow of p38a-DMC mice compared to WT mice (Fig EV4F and 4G). Taking into consideration that IGF-1-DMC mice also exhibit reduced Ly6ChiCCR2+ monocytes in the bone marrow under homeostatic conditions (Fig EV4B), our results indicate that the effect on bone marrow monocytes is directly caused by IGF-1. This is further supported by the observation that myeloid cells sorted from the bone marrow (including monocytes and macrophages) show LysM-Cre activity (Fig EV54E).

3) The chemokines are already reduced in baseline colon and macrophages. However, the number of F4/80+ cells in healthy colon does not seem to be changed (Figs 1F, 2C). Can this be really the cause for the reduced migration of inflammatory monocytes to the colon? How are chemokine levels in early DSS treated mice? What about other important inflammatory mediators in DSS colitis, such as the Th17 inflammatory response? Otsuka et. al have previously reported reduced levels of important inflammatory cytokines (e.g. IL6, IL12, Il1b, TNF etc.) in p38dMC mice after DSS colitis. How does this fit into the overall picture proposed here? There is no mention in the proposed model or discussion.

This is an important observation. Indeed, the reduction of chemokines in the colon and inflammatory monocytes in the bone marrow did not correlate with the percentage of F4/80+ macrophages. Therefore, we quantified the amount of myeloid cells in the colon under basal conditions and found reduced numbers of CD45+CD11b+ cells in p38a-DMC mice compared to WT mice (Fig EV4D). We can think of two possible explanations. One is that although recruitment of monocytes to the colon is impaired in p38a-DMC mice compared to WT mice, these cells do not (yet) express F4/80 under basal conditions in the colon. Of note, although monocytes also express F4/80 at low or intermediate levels, these cells acquire the expression of the F4/80 antigen particularly during the process of differentiating into macrophages (Bain, Scott et al., 2013). The second possibility might be that the percentage of F4/80+ macrophages under basal conditions is very low (see Fig 1E, 2C, 5H and Appendix Fig S1F and S4D) and taking into account the variability between mice it is very difficult to reach statistical significance, even with a high number of colons analyzed.

As the reviewer suggested, we have now analyzed several chemokines and cytokines in DSS treated mice (Fig 2E and Appendix Fig S2G). From these analysis we could identify several cytokines and chemokines that, as expected, were reduced in whole colon extracts derived from DSS-treated p38a-DMC mice compared to WT. Moreover, at this time point when inflammation triggers the pathological expression of these molecules and the recruitment of monocytes to the colon, we detected significant differences in F4/80+ cells in the colons of WT and p38a-DMC mice (Fig 2C, 5H and Appendix Fig S4D). We observed a strong downregulation of IL-17 (Fig 2E and Appendix Fig S2G), which is in line with the observation that STAT3 activation (important regulator of the TH17 response) (Rebe, Vegran et al., 2013) was downregulated in p38a-DMC mice. As previously reported by Otsuka et al. (2010), we also observed downregulation of IL-1b and TNF-a (Fig 2E and Appendix Fig S2F and S2G).

4) *The authors show that there are reduced numbers of wound-healing macrophages (and reduced IGF1) in p38dMC mice after DSS colitis. However, epithelial regeneration is not affected and seems to be even better compared to wt mice. How do the authors explain/discuss these findings?*

Our analysis shows a significant reduction of macrophages at day 7 of DSS treatment. However, at this stage (the acute pro-inflammatory phase), these macrophages are still rather pro-inflammatory than anti-inflammatory. This correlates also with the reduction of the proinflammatory cytokines IL-1b and TNF-a previously described by Otsuka et al. (2010) to be downregulated in the p38a-DMC mice compared to the WT. As we mention in the Discussion (page 12 and 13) that the recruitment of leukocytes (including monocytes and macrophages) is associated with inflammation and tissue damage due to the secretion of pro-inflammatory cytokines. Indeed, the reduction of IGF-1 (marker for wound-healing macrophages), becomes clearer in the anti-inflammatory repair phase (day 13). Taking into consideration that p38a also regulates other cytokines than IGF-1, which result in stronger immune cell recruitment and tissue damage, we conclude that it is difficult to compare regeneration when it originates from different extents of initial tissue damage. We further discuss the discrepancies at day 13 (in the repair phase) between the p38a-DMC mice and the IGF-1-DMC mice in the Discussion (pages 11 and 12): “*However, in the repair phase, no differences in epithelial damage could be observed between DSS-treated WT and IGF-1-DMC mice. Notably, at this stage, inflammatory cell recruitment is not as prominent as in the acute pro-inflammatory phase, due to the withdrawal of DSS and the onset of repair mechanisms. We therefore hypothesize that IGF-1 contributes to a more efficient repair in WT mice, resulting in similar levels of epithelial damage compared with IGF-1-DMC mice. We conclude that p38a-DMC and IGF-1-DMC mice do not phenocopy each other due to both the residual levels of IGF-1 signaling present in p38a-DMC mice and the regulation by p38a of other inflammatory mediators.*”

5) *Showing IGF1 / p38 stainings in CRC and/or colitis/IBD patients would have strengthen the medical relevance of the study.*

We have analyzed human samples from 23 colons of ulcerative colitis patients and from 25 colon tumors. Interestingly, this analysis showed a significant correlation between the activation of the IGF-1 pathway and the presence of macrophages with p38 activity. Of note, we could not observe any correlation in samples derived from control individuals, suffering neither IBD nor colon cancer (Fig 8).

6) *How is phospho pSTAT3 quantified? It would make sense to delineate staining on epithelial vs stromal cells? Can pSTAT3 be directly induced by IGF1 in colonocytes or myeloid cells? How is pSTAT3 in IGF1 dMC mice?*

As shown in Figure 1E and explained in “Materials and methods” (page 19), we focused on the mucosal layer for the immunohistochemistry analysis of stained colons. As suggested by the reviewer, we have also performed double stainings with phospho-STAT3 and E-Cadherin, as a marker for epithelial cells. However, we found a dramatic reduction in E-Cadherin expression in damaged tissue, which we did not expect. Afterwards, we realized that this has been described by other groups and that this is observed in patients with ulcerative colitis (Eichele & Kharbanda, 2017, Wang, Zhuang et al., 2014). However, given that loss of ECadherin staining is associated with epithelial damage, we could further confirm the histopathological evaluation of epithelial damage, since colons from WT mice expressed significantly less E-Cadherin compared to p38a-DMC mice. Nonetheless, the regions analyzed were limited to the mucosal layer, excluding muscular and serosa layer, therefore the majority of cells analyzed are epithelial cells, although we cannot exclude that infiltrating stromal cells might also contribute. In this line, we found several reports proposing that STAT3 phosphorylation can be induced by IGF-1 in various cell types including colonocytes (Flashner-Abramson, Klein et al., 2016, Sanchez-Lopez, Flashner-Abramson et al., 2016, Xu, Zhou et al., 2017, Yao, Su et al., 2016, Zhang, Zong et al., 2006, Zong, Chan et al., 2000).

However, we are not aware of reports indicating that IGF-1 can induce STAT3 phosphorylation in myeloid cells. Furthermore, we observed reduced levels of STAT3 phosphorylation in the colons of DSS-treated IGF-1-DMC mice compared to WT mice (Fig EV3E). Interestingly, treatment with IGF-1 impaired the reduction of STAT3 phosphorylation levels observed in DSS-treated p38a-DMC mice compared to WT mice (Fig 6D), suggesting that, directly or indirectly, IGF-1 controls STAT3 phosphorylation levels.

7) Figure 1a and 4b look identical, as well as Suppl. Fig. 4E and F. Please clarify.

We thank the reviewer for their attention to detail and apologize for not having noticed that both figures showed the same panels in our original submission. This has now been corrected in the new Figures 1A and 4B, and Figures EV3E and 3F.

8) F4/80 staining in healthy colon is generally weak. Only 1 or 2 cells / section in healthy colon is very few. Did the staining work properly?

The staining for F4/80 is a standard procedure set up and performed in the histopathology facility of our Institute. In our opinion, the staining is very specific with almost no background. However, we would not expect to have a high number of F4/80+ cells in untreated mice. Moreover, the untreated control samples were stained in parallel with the samples treated with DSS or AOM/DSS, which worked nicely and showed the expected increased number of macrophages, so we think it unlikely that the staining did not work properly only in control samples. It is possible that the number of F4/80+ cells in colons from untreated mice will be affected by the mouse housing facilities.

9) What does % of cells mean (e.g. Fig. 1E,F)? Are Serosa cells also included in counting? Should the sections rather be counted per colon area?

We have added a sentence to the legend of Figure 1E (new Fig 1D) to explain how these cells were quantified. Basically, the percentage of Ly6ChiCCR2+ cells indicated in the figure refers to the bone marrow cells that were alive and CD45+CD11b+. Moreover, as explained above (point 6), we have added a sentence in Materials and Methods (page 19) and the legend to Figure 1E explaining how the colon immunohistochemistry stainings were analyzed. Serosa cells were not included in the counting, therefore, although we are aware that analyzing per area is another way to perform these quantifications, we do not think that this should modify our results and the conclusions drawn. We calculated the results of all our immunohistochemistry stainings by taking into account the total number of cells analyzed rather than the area, since the macros that we used for the quantifications worked very nice in this way. Additionally, the TMarker software that we have used for analyzing the different phospho-IGF1R intensities does not analyze per area but refers to the percentage of stained cells versus the total number of cells analyzed. Therefore, we thought that it would be more consistent to always represent our data as the percentage of positive cells versus the total number of cells analyzed.

10) The following pictures are too small and higher magnifications should be shown: Fig. 1F, Fig. 2D-F

We now show higher magnifications in Appendix Figure S1F and S1C-S1E.

11) The reference to Suppl. Fig1A in the result and the respective Figure do not match.

We apologize for the possible confusion in the text and have corrected the phrase in the Results section (page 4).

Reviewer #2

In this manuscript, Youssif et al examine the role of myeloid p38a signaling in a model of colitis and colitis-associated tumorigenesis. Contrary to epithelial p38a signaling which is tumor suppressive during tumor initiation, now authors report that myeloid p38a signaling has a tumor promoting function. Accordingly, epithelial damage and inflammation was reduced in p38aDMC mice. Moreover, they suggest that p38a in myeloid cells controls IGF1 expression, which acts as chemo-attractant for inflammatory cells, and seems important for colitis-associated tumorigenesis. Blocking IGF1 signaling using chemical inhibitor reduced epithelial damage and tumor load. The p38a-IGF1 link during colon tumorigenesis is interesting but there are several concerns:

1. The phenotype that p38a depletion in myeloid cells reduces DSS induced epithelial damage and impairs number of inflammatory cells in the mucosa was already published, as author also cited that work. However, the link between p38a and IGF1 signaling in this context and its impact on tumorigenesis is new. Nevertheless, the suggested mechanism is not entirely convincing. If the

authors propose that reduced myeloid cell recruitment is responsible for improved outcome of DSS colitis, this cannot entirely explain the phenotype as it has been shown that macrophages are required for proper wound healing. In particular Fig. 4C is not very convincing in this context. It is not clear if the proliferation rate shown is referring to epithelial cells. As the histology in the 7 day DSS panels does not show any epithelial cells but only a big ulcerations. Seeing this it also very surprising to note that at 13 days the mucosa in the knockout samples is completely healed whereas there is still massive inflammation in the wt mice. Are these really representative stainings? A lower magnifications should be provided to see the entire distal colon of the mice analyzed. In addition to the clinical data (DAI) the authors should provide a rigorous histological assessment of the histological damage at both time points. Weight loss and tissue damage does not necessarily correlate between male and female mice. Female mice are more resistant to weight loss, yet they can show the same histological damage as male mice. If wound healing/regeneration is impaired in ko mice, initial tissue damage should be less otherwise mice would show more weight loss and not less.

We thank the reviewer for the positive appreciation of our work and for their constructive criticisms. As the reviewer correctly mentions, macrophages are required for proper wound healing, but macrophages in p38a-DMC mice do not completely disappear, they are solely reduced during DSS-induced colitis, reaching statistical significance at day 7 in the acute inflammatory phase (Fig 2C, 5H and Appendix Fig S4D). In this context, it is commonly accepted that macrophages in the colons promote tissue damage and inflammation (Bain & Mowat, 2014b). Moreover, the inhibition of macrophage recruitment to the colon by disruption of the CCL2-CCR2 chemokine axis, markedly reduces circulating monocytes and ameliorates acute intestinal inflammation (Bain & Mowat, 2014a, Ginhoux & Jung, 2014).

Thus, increased leukocyte infiltration is considered to be a hallmark of IBD and experimental colitis, contributing to disease initiation and tissue damage (Abraham & Cho, 2009). As commented in point 6 above, the regions analyzed for all of the colon immunohistochemistry stainings, including the Ki67 staining, were limited to the mucosal layer, excluding muscular and serosa layers. Therefore, the majority of cells analyzed are epithelial cells, although we cannot exclude that infiltrating stromal cells might have a small effect on the observed results.

As mentioned in the Results section (page 7), we found significant differences in cell proliferation, as determined by Ki67 staining, between the colons of DSS-treated WT and p38a-DMC mice in the repair phase at day 13 (Fig 4C and Appendix Fig S3C), when IGF-1 protein levels were significantly different in colon extracts (Fig 4A). This might be related to the stronger tissue damage in the WT mice, inducing stronger repair and proliferation mechanisms, or it might be related to the increased IGF-1 levels, in accordance with the known mitogenic properties of IGF-1. Injury and ulceration induce wound-healing/regeneration responses, which includes migration of stem cells and their enhanced proliferation and expansion to fill in for damaged mucosa (Kuraishy, Karin et al., 2011).

However, cell migration and proliferation, apart from being important for wound-healing processes, are also pro-tumorigenic properties of cancer cells (Schafer & Werner, 2008). If these cells harbor oncogenic mutations, which might be induced by the pro-inflammatory environment, local repetitive injury and regeneration will instigate their proliferation and tumor formation (Kuraishy et al., 2011). We have histologically assessed the epithelial damage at both time points in the two genetic mouse models and in response to the different treatments (Fig 2B, 5B, 5F, 6C, 6D, EV2A, EV3D and Appendix FigS4C and S4F), which is now described in Materials and Methods (pages 20 and 21). We agree with the reviewer, that the image previously used to illustrate the Ki67 staining at day 13 in p38a-DMC mice was not the most representative in terms of tissue damage, and have now replaced it by a more representative one (Fig 4C). Moreover, as suggested by the reviewer, we provided lower magnifications (Appendix Fig S3C) to see the entire distal colon of the mice analyzed.

Finally, as the reviewer also mentions, wound healing/regeneration (proliferation) is impaired in p38a-DMC mice in which there is reduced initial tissue damage, as these mice also show less body weight loss and disease activity index compared to WT mice.

2. In tumors no difference was observed in tumor size between Wt and p38a KO mice (Suppl Fig1D). However, at this point both phospho-IGF1R and Phospho-Stat3 seems to be more in WT

tumors compared to p38aKO tumors. As authors wrote in the text, that Stat3 and IGF1 signaling both are important for cell proliferation and considering the reduced Stat3 and IGF1 signaling in KO tumors, one would expect to have less proliferation in tumors and perhaps also smaller tumors. Proliferation and apoptosis rates need to be determined in these tumors.

We thank the reviewer for this suggestion, which has prompted us to re-evaluate the tumor sizes in animals from both genetic models. We realized that although the average tumor size was not significantly different (Fig EV1C for p38a-DMC mice and Fig 7A for IGF-1-DMC mice), the number of large tumors in the WT mice was significantly increased compared to both knockout models (Fig EV1D and Fig 7B). Moreover, we observed higher proliferation rates evaluated by Ki67 staining in tumors derived from p38a-DMC mice compared to the WT mice (Fig EV1E). However, apoptosis levels determined by TUNEL or cleaved Caspase 3 staining did not show significant differences between the tumors from WT and p38a-DMC mice (Appendix Fig S1C and S1D). Therefore, as suggested by the reviewer, we conclude that the increased tumorigenesis observed in WT mice is most probably due an increased proliferation rate, driven by IGF-1 and STAT3 signaling, as observed in the repair phase after DSS treatment (Fig 4C and Appendix Fig S3C). However, it should be noted that although WT mice show more big tumors than the p38a-DMC mice, the increased tumor burden in WT mice is mainly due to an increased number of tumors, meaning that tumor initiation is probably impaired in p38a-DMC mice. This was also observed at earlier stages of tumorigenesis (day 55), where WT mice already exhibit more tumors than p38a-DMC mice (Fig 7E). Therefore, it is expected that tumors that initiate earlier, will also be of larger size at the end of the experiment, even if proliferation and/or apoptosis rates are similar.

3. Differences in the cytokines production (IL1b and TNFa, Fig 2F) upon DSS treatment are very moderate. Moreover, it was shown that IL1b KO mice are more susceptible to DSS induced colitis (PMID: 23793223), which is somewhat contradictory to what is observed in p38a KO mice. Perhaps these differences can be attributed to other cytokines and chemokines regulated by p38a.

IL1b and TNFa are usually associated directly with NFkB signaling (also shown by PMID 20080092), rather than Stat3 signaling which could be regulated by these cytokines indirectly. Author should also examine IL6 family cytokines, which are direct inducers of Stat3 signaling.

We agree with the reviewer and also comment in the Discussion section (page 12) that p38a is known to regulate several inflammatory cytokines. Moreover, we have now performed a cytokine array (Fig 2E and Appendix Fig S2G), which confirmed the results from the ELISAs, and we also observed the downregulation of additional pro-inflammatory cytokines and chemokines. We agree that the changes in TNFa expression that we observed are not very impressive, nevertheless there was a reduction using both ELISA and cytokine arrays. However, IL-1b was significantly reduced in the results from the ELISA and could also be confirmed in the cytokine array.

The role of IL-1b in intestinal inflammation seems to be complex, which might contribute to some conflicting reports. IL-1 β levels in the colons of patients with IBD correlate with disease activity and high levels of IL-1 β were associated with active lesions (Cominelli & Pizarro, 1996, Ludwiczek, Vannier et al., 2004), suggesting an important role of this cytokine in promoting localized inflammation. High levels of colonic IL-1 β are also a feature of several animal models of colitis (Cominelli, Nast et al., 1990, Okayasu, Hatakeyama et al., 1990). The importance of IL-1 β in modulating intestinal inflammation has been confirmed by infection studies, as blocking IL-1 β ameliorated inflammatory pathology in both *Clostridium difficile*-associated colitis and *Salmonella typhimurium*-induced enteritis (Muller, Hoffmann et al., 2009, Ng, Hirota et al., 2010). Moreover, IL-1b can also activate the release of other pro-inflammatory cytokines such as TNF-a, IL-23 and IL-6 (Sahoo, Ceballos-Olvera et al., 2011, Tsianos & Katsanos, 2009). We also evaluated IL-6 family cytokines in the cytokine array. Although IL-6 was almost undetectable in these samples, the expression of other activators of STAT3, such as G-CSF and IL-27 (Rebe et al., 2013) was reduced in early colitis (Fig 2E and Appendix Fig S2G) as well as in the tumors (Fig 1F and Appendix Fig S1G).

4. Regulation of IGF1 by p38a in BMDMs is very clear as shown in Fig 3 but the differences in IGF1 levels in vivo during colitis are very mild. Similarly, P-IGF1R staining in the p38a KO tumors has slightly reduced expression compared to WT tumors. Authors should also measure IGF1 in the tumors.

We think that the results obtained using p38a-DMC BMDMs *in vitro* (Fig 3A) are consistent with those obtained *in vivo*. However, the use of p38 chemical inhibitors in BMDMs results in stronger downregulation (Fig 3B and 3C), probably due to both an effective inhibition of p38a by the chemical inhibitors, as well as the fact that the compounds used also inhibit p38b, which might contribute. Of note, it is unusual to see total downregulation of p38a in LysM-Cre mice (Fig 1A and 5A), which probably also results in less impressive IGF-1 downregulation. As suggested by the reviewer, we have now measured IGF-1 levels in the tumors from p38a-DMC mice and WT mice, but the changes observed were small and did not reach statistical significance. Similarly, the downregulation of IGF-1 in tumors derived from IGF-1-DMC mice was moderate. We therefore think that either a small, local reduction of IGF-1 could affect tumorigenesis in mice, or that our observations rather reflect the effect of IGF-1 at earlier stages of tumorigenesis, consistent with the importance of this factor in tumor initiation (Clayton, Banerjee et al., 2011, Yu & Rohan, 2000).

5. In the entire manuscript, number of mice used in the experiments is mentioned but how many times experiments were repeated, is not mentioned.

We have added the missing information to the corresponding figures.

6. *Suppl Fig 4B and C: Effect of IGF1 depletion in myeloid cells has very little effect on body weight loss and DAI compared to p38a KO mice. 31 mice are used for DAI quantification, but still differences between groups are not impressive. Is this pool data or from one single experiment? If it's a pool, then authors should show individual experiments.*

In fact, this was a pool from three experiments. These are the only data that were pooled in the entire manuscript, since the differences were moderate in the three individual experiments. Nevertheless, the three experiments showed a similar tendency. We have now indicated this in the figure legend and also show the data of the individual experiments (Appendix Fig S3E).

7. *Compared to p38a KO mice, the differences in phospho-Stat3 staining in IGF1 KO mice does not seem significant. How is Stat3 signaling regulated in these experiments? Similarly, IGF1 KO mice do not have (or very little) any effect on P-IGF1R as shown in Fig S4.*

We thank the reviewer for this observation, which prompted us to analyze more IGF-1-DMC mice, to evaluate whether the changes observed in the phosphorylation of STAT3 and IGF-1 receptor were significant. Indeed, we now show significant differences between WT and IGF-1-DMC mice for p-STAT3 at day 7 and for p-IGF1R at day 7 and day 13 (Fig EV3E and 3F).

As commented above (point 6 of Reviewer 1), it is possible that IGF-1 signaling directly activates STAT3 in these experiments, as proposed in previous reports (Flashner-Abramson et al., 2016, Sanchez-Lopez et al., 2016, Xu et al., 2017, Yao et al., 2016, Zhang et al., 2006, Zong et al., 2000). Nevertheless, we think that STAT3 phosphorylation might be also indirectly induced by other STAT3 activating cytokines secreted by infiltrating immune cells, given that IGF-1 acts as a chemokine and increases immune cell recruitment to the colons (Fig 5C, 5D, 5G, 5H, 6B, 7C and Appendix Fig S4D). We have now elaborated on these observation in the Discussion (page 13).

8. *IGF1 KO mice have reduced tumor load without affecting the size. In fig 6A, 22 mice were used. Is this pooled data? If not, shown individual experiments. Interestingly, F4/80+ cells were significantly reduced in tumors from IGF1 KO mice but was not in non-tumoral area of these mice despite having similar percentage of F4/80+ cells in tumors vs non-tumor. Discuss this tumor specific effect.*

In fact, as commented above (point 2 of Reviewer 1), we realized that although the average tumor size was not significantly different (Fig EV1C for the p38a-DMC mice and Fig 7A for the IGF-1-DMC mice) the number of large tumors was significantly increased in the WT mice compared to both knockout models (Fig EV1D and Fig 7B). This is not pooled data, all the mice were analyzed from the same experiment. As stated by the reviewer, we actually observed a tumor specific effect of reduced F4/80+ cells in both knockout models (Fig 1E and 7C). We therefore analyzed a cytokine array comparing tumors derived from WT and p38a-DMC mice (Fig 1F and Appendix Fig S1G), and observed reduced levels of several chemokines in tumors from the p38a-DMC mice, which could be implicated in the recruitment of F4/80+ cells to the tumors. Taking into account that IGF-1

is likely to act as a chemokine in this model, we hypothesize that this tumor specific effect of macrophage recruitment is mainly due to IGF-1 secreted by myeloid cells in the tumors.

9. Mechanistic details how p38a-IGF1 signaling axis affect epithelial repair and tumorigenesis is not fully delineated. Recently it was shown that circulating IGF1 can control colonic stem cell function in diabetic enteropathy (PMID: 26431183). Authors should check number of stem cells during DSS-induced colitis, repair and tumors both in p38a and IGF1 KO mice. In addition, 3D organoids from colon should be used to functionally examine the role of IGF1 on stem cells as this modes can serve as a toll to examine regeneration.

We thank the reviewer for this suggestion. However, as commented above (point 2 of Reviewer 1), we could not detect differences in serum IGF-1 levels between WT and p38a-DMC mice (Appendix Fig S3B), most probably because the major source of circulating IGF-1 is the liver and not myeloid cells (Gow et al., 2010). We also analyzed the expression of several genes associated to stem cells and differentiation, such as *Lgr5*, *muc2*, *hopx*, *sox9*, *Lgr1*, Lysozyme and *Neurog3*, but did not detect any significant changes between WT and p38a-DMC mice. Moreover, Ephrin B2 and aldehyde dehydrogenase were also evaluated by immunohistochemistry without detecting significant differences during colitis or tumorigenesis. We have no evidence at the moment that colonic stem cell function is affected in our model, and did not further pursue this. However, we have been able to analyze colon tissues from ulcerative colitis patients as well as colon tumors, and found a positive correlation between the phosphorylation of IGF-1 receptor and the presence of macrophages with phosphorylated p38a, supporting the potential clinical relevance of the p38a-IGF-1 axis in human disease (Fig 8).

10. In Fig1B, statistical data on rectal prolapse comparison between wt and p38a ko mice should be provided

We now show a histogram with the percentages calculated from two experiments (Appendix Fig S1B). We have to recognize that although we consistently observed a reduced percentage of p38a-DMC mice with anal prolapse, when combining the two experiments in which this data was recorded, the presence of anal prolapse in WT and p38a-DMC mice was not statistically different. These two groups of mice were analyzed in three tumorigenesis experiments, but in the first experiment the anal prolapse frequency was not recorded.

References

- Abraham C, Cho JH (2009) Inflammatory bowel disease. *N Engl J Med* 361: 2066-78
- Bain CC, Mowat AM (2014a) Macrophages in intestinal homeostasis and inflammation. *Immunol Rev* 260: 102-17
- Bain CC, Mowat AM (2014b) The monocyte-macrophage axis in the intestine. *Cell Immunol* 291: 41-8
- Bain CC, Scott CL, Uronen-Hansson H, Gudjonsson S, Jansson O, Grip O, Williams M, Malissen B, Agace WW, Mowat AM (2013) Resident and pro-inflammatory macrophages in the colon represent alternative context-dependent fates of the same Ly6Chi monocyte precursors. *Mucosal Immunol* 6: 498-510
- Clayton PE, Banerjee I, Murray PG, Renehan AG (2011) Growth hormone, the insulin-like growth factor axis, insulin and cancer risk. *Nat Rev Endocrinol* 7: 11-24
- Cominelli F, Nast CC, Clark BD, Schindler R, Lierena R, Eysselein VE, Thompson RC, Dinarello CA (1990) Interleukin 1 (IL-1) gene expression, synthesis, and effect of specific IL-1 receptor blockade in rabbit immune complex colitis. *J Clin Invest* 86: 972-80
- Cominelli F, Pizarro TT (1996) Interleukin-1 and interleukin-1 receptor antagonist in inflammatory bowel disease. *Aliment Pharmacol Ther* 10 Suppl 2: 49-53; discussion 54
- Eichele DD, Kharbanda KK (2017) Dextran sodium sulfate colitis murine model: An indispensable tool for advancing our understanding of inflammatory bowel diseases pathogenesis. *World J Gastroenterol* 23: 6016-6029
- Flashner-Abramson E, Klein S, Mullin G, Shoshan E, Song R, Shir A, Langut Y, Bar-Eli M, Reuveni H, Levitzki A (2016) Targeting melanoma with NT157 by blocking Stat3 and IGF1R signaling. *Oncogene* 35: 2675-80
- Ginhoux F, Jung S (2014) Monocytes and macrophages: developmental pathways and tissue homeostasis. *Nat Rev Immunol* 14: 392-404
- Gow DJ, Sester DP, Hume DA (2010) CSF-1, IGF-1, and the control of postnatal growth and

- development. *J Leukoc Biol* 88: 475-81
- Kuraishy A, Karin M, Grivennikov SI (2011) Tumor promotion via injury- and death-induced inflammation. *Immunity* 35: 467-77
- Ludwiczek O, Vannier E, Borggraefe I, Kaser A, Siegmund B, Dinarello CA, Tilg H (2004) Imbalance between interleukin-1 agonists and antagonists: relationship to severity of inflammatory bowel disease. *Clin Exp Immunol* 138: 323-9
- Muller AJ, Hoffmann C, Galle M, Van Den Broeke A, Heikenwalder M, Falter L, Misselwitz B, Kremer M, Beyaert R, Hardt WD (2009) The S. Typhimurium effector SopE induces caspase-1 activation in stromal cells to initiate gut inflammation. *Cell Host Microbe* 6: 125-36
- Ng J, Hirota SA, Gross O, Li Y, Ulke-Lemee A, Potentier MS, Schenck LP, Vilaysane A, Seamone ME, Feng H, Armstrong GD, Tschopp J, Macdonald JA, Muruve DA, Beck PL (2010) Clostridium difficile toxin-induced inflammation and intestinal injury are mediated by the inflammasome. *Gastroenterology* 139: 542-52, 552 e1-3
- Okayasu I, Hatakeyama S, Yamada M, Ohkusa T, Inagaki Y, Nakaya R (1990) A novel method in the induction of reliable experimental acute and chronic ulcerative colitis in mice. *Gastroenterology* 98: 694-702
- Rebe C, Vegran F, Berger H, Ghiringhelli F (2013) STAT3 activation: A key factor in tumor immunoescape. *JAKSTAT* 2: e23010
- Sahoo M, Ceballos-Olvera I, del Barrio L, Re F (2011) Role of the inflammasome, IL-1beta, and IL-18 in bacterial infections. *ScientificWorldJournal* 11: 2037-50
- Sanchez-Lopez E, Flashner-Abramson E, Shalapour S, Zhong Z, Taniguchi K, Levitzki A, Karin M (2016) Targeting colorectal cancer via its microenvironment by inhibiting IGF-1 receptor-insulin receptor substrate and STAT3 signaling. *Oncogene* 35: 2634-44
- Schafer M, Werner S (2008) Cancer as an overhealing wound: an old hypothesis revisited. *Nat Rev Mol Cell Biol* 9: 628-38
- Tsianos EV, Katsanos K (2009) Do we really understand what the immunological disturbances in inflammatory bowel disease mean? *World J Gastroenterol* 15: 521-5
- Wang B, Zhuang X, Deng ZB, Jiang H, Mu J, Wang Q, Xiang X, Guo H, Zhang L, Dryden G, Yan J, Miller D, Zhang HG (2014) Targeted drug delivery to intestinal macrophages by bioactive nanovesicles released from grapefruit. *Mol Ther* 22: 522-534
- Xu L, Zhou R, Yuan L, Wang S, Li X, Ma H, Zhou M, Pan C, Zhang J, Huang N, Shi M, Bin J, Liao Y, Liao W (2017) IGF1/IGF1R/STAT3 signaling-inducible IFITM2 promotes gastric cancer growth and metastasis. *Cancer Lett* 393: 76-85

2nd Editorial Decision

07 May 2018

Thank you for the submission of your revised manuscript to EMBO Molecular Medicine. We have now received the enclosed reports from the referees that were asked to re-assess it. As you will see, the reviewers are now globally supportive and I am pleased to inform you that we will be able to accept your manuscript pending the following final amendments:

- 1) Please address the last set of concerns commented by referee 1.

Please address the referee's comments in writing. At this stage, we'd like you to discuss the points raised and if you do have data at hand, we'd be happy for you to include it, however we will not ask you to provide any additional experiments at this stage.

Please provide a letter INCLUDING all my comments and the reviewer's reports and your detailed responses to their comments (as Word file).

Please submit your revised manuscript within two weeks. I look forward to seeing a revised form of your manuscript as soon as possible.

***** Reviewer's comments *****

Referee #1 (Comments on Novelty/Model System for Author):

The model used is state of the art

Referee #1 (Remarks for Author):

The authors have addressed and clarified the concerns raised in the previous submission. The paper has greatly improved and the overall conclusion is significantly strengthened by including additional data addressing the mechanistic link between p38/IGF1.

Referee #2 (Comments on Novelty/Model System for Author):

- 1) Experiments well performed and controlled
- 2) There is a manuscript describing myeloid-specific deletion of p38, which has however not described IGF1 as a target.
- 3) Inclusion of human data has improved this point

Referee #2 (Remarks for Author):

In the revised manuscript by Youssif and colleagues the authors now provide new data from ulcerative colitis patients and human colon tumors and show a positive correlation between IGF1R phosphorylation and presence of macrophages with phosphorylated p38a supporting the clinical significance of p38a-IGF1 axis in these pathological conditions. Most of the comments were addressed adequately by acquiring new data therefore manuscript has significantly improved. My only remaining concerns are as follows:

- 1) Authors show tumor specific effect of reduced F4/80+ cells in both knockout models which is now supported by reduced expression of cytokines/chemokines in the tumors from p38a KO mice. In response to comment 4, authors quantified IGF-1 levels from both KO tumor models, but changes were not significant (data not shown) which suggest that IGF-1 within the tumor microenvironment does not play a role in the recruitment of macrophages. If IGF1 secreted by myeloid cells plays a role in macrophage recruitment then there should be less number of macrophages in AOM/DSS epithelium as shown in IGF-1 KO tumors (Fig 7C). Similar to cytokine/chemokine array done in the tumors from p38a KO mice, authors should analyze these in the AOM/DSS epithelium vs tumors from IGF-1 ko mice. If there are changes in chemokines/cytokines then author should discuss how they might be regulated by IGF-1.
- 2) During repair phase at day13, there is no difference in epithelial damage in IGF-1 KO mice but still reduced phospho-Stat3 and Phospho-IGF1R staining in these mice. Does this correlate with proliferation at this time point?
- 3) Does secreted IGF-1 activate p38a signaling in myeloid cells and/or in epithelial cells as a feedback during DSS and tumorigenesis?
- 4) Chemokine/cytokine arrays were performed from total colon lysates from DSS-treated or colon tumors from WT and p38aKO mice. Although it is known that p38a can regulate several of these mediators, one should be cautious in interpreting data as these differences can also be due to different number of immune cells/epithelial cells present in p38a KO vs WT mice.
- 5) Fig 7C, p38a-DMC should be correctly labeled as IGF-1-DMc.

2nd Revision - authors' response

16 May 2018

Authors made requested editorial changes.

Corresponding Author Name: Angel R. Nebreda
Journal Submitted to: EMBO Molecular Medicine
Manuscript Number: EMM-2017-08403